# A Structure from Motion photogrammetry-based method to generate sub-millimetre resolution Digital Elevation Models for investigating rock breakdown features

Ankit K. Verma[1] and Mary C. Bourke[1]

[1]Department of Geography, Trinity College Dublin, The University of Dublin, Dublin, Ireland

*Correspondence to:* Ankit K. Verma (vermaan@tcd.ie)

## Abstract

We have generated sub-millimetre resolution DEMs of weathered rock surfaces using SfM photogrammetry techniques. We apply a close-range Structure from Motion (SfM) photogrammetry-based method in the field and use it to generate high-resolution topographic data for weathered boulders and bedrock. The method was pilot tested on extensively weathered Triassic Moenkopi Sandstone outcrops near Meteor Crater in Arizona. Images were taken in the field using a consumer grade DSLR camera and were processed in commercially available software to build dense point clouds. The point clouds were registered to a local 3D coordinate system (x, y, z) which was developed using a specially designed triangle coded control target and then exported as Digital Elevation Models (DEMs). The accuracy of the DEMs was validated under controlled experimental conditions. A number of checkpoints were used to calculate errors. We also evaluated the effects of image and camera parameters on the accuracy of our DEMs. We report a horizontal error of 0.5 mm and vertical error of 0.3 mm in our experiments. Our approach provides a low-cost method, for obtaining very high-resolution topographic data on weathered rock surfaces (area < 10 m$^2$). The results from our case study confirm the efficacy of the method at this scale and show that the data acquisition equipment is sufficiently robust and portable. This is particularly important for field conditions in remote locations or steep terrain where portable and efficient methods are required.

## Keywords

Rock breakdown, Geomorphology, Structure from Motion, Close-range photogrammetry, Digital Elevation Model, Micro-topographic survey

## 1. Introduction

Rock breakdown describes a range of geomorphic processes that transform rock masses into soil or regolith and unconsolidated rock materials. It plays a vital role in climate control via atmosphere-lithosphere interaction, biogeochemical cycling and landform evolution on a planetary scale (Goudie and Viles, 2012). The scale of features range from µm (e.g., fractures, weathering pits, fractures) to m scale (e.g., tafoni, scaling and blisters) (Viles, 2001;Bourke and Viles, 2007). In addition, many active rock breakdown processes that operate over a short geological timescale ($10^0$-$10^2$ years) produce observable microscale (mm-cm) breakdown features. To better understand the weathering processes, high resolution (sub-mm to mm) microtopographic data are necessary for in-situ measurement of small-scale weathering features (Viles, 2001). To date, the inability to measure the general geomorphometry of small-scale breakdown features has inhibited our understanding of the causal links at relevant scales. Many small-scale (mm-cm) breakdown features are ambiguous, and it remains challenging to distinguish between similar looking features (e.g. aeolian pits vs dissolution pits) and therefore to establish a clear link between weathering feature form and the formative process. Even for homogenous forms on a surface, it may be difficult to understand the role of individual weathering mechanisms (Viles, 2005;Warke, 2007;Viles, 2010;Viles et al., 2018). In addition, extending analysis routines between rock breakdown sites, to better understand features that often show considerable complexity in their intensity, size and shape depending on lithological, geological and micro-environmental factors (Viles, 2001) has been limited by the application of different techniques at different scales and in different locations. Using the same technique (i.e. SfM) across scales will permit similar analysis routines for different scale landscapes (Cullen et al., 2018).

This will facilitate the investigation of potential feedbacks across various scales boundaries. The morphometric analysis of topography at different scales will aid interpretation of the complex interrelationship of weathering processes and landscapes and facilitate a better understanding of the multi-scale weathering system (Viles, 2013).

Quantitative analysis of landforms is necessary for the identification and interpretation of landform genesis and history. In the past few decades, a range of micro-topographic data collection methods have been used in rock breakdown and soil erosion studies. These include: (1) laser scanning techniques (Fardin et al., 2001;Fardin et al., 2004;Bourke et al., 2007;Bourke et al., 2008;Aguilar et al., 2009;Sturzenegger and Stead, 2009;MŁynarczuk, 2010;Medapati et al., 2013;Chen et al., 2014;Ge et al., 2014;Lai et al., 2014), (2) stereophotogrammetry (Rieke-Zapp and Nearing, 2005;Taconet and Ciarletti, 2007;Aguilar et al., 2009;Bui et al., 2009;Sturzenegger and Stead, 2009;Kim et al., 2015), (3) Micro-roughness meters (MRM) (McCarroll, 1992;McCarroll and Nesje, 1996;White et al., 1998). However, there are significant logistical, technical and for some, financial constraints that have hindered the adoption of these methods, particularly in physically challenging terrains such as remote, difficult to access and steep terrains.

Laser scanning permits collection of high-resolution topographic data at the relevant scale for the study of small-scale rock breakdown features. However, due to difficulties associated with transporting the often-cumbersome instrument in the field (Ehlmann et al., 2008), this technology has rarely been used to collect data on rock surfaces in situ (Fardin et al., 2004). Additionally, laser scanners require a stable platform, on which to operate and this can be difficult to find in steep terrain (e.g. crater and canyon walls, and mountainous terrain). There are hand-

 held portable laser scanners available which do not require a stable platform to operate, but the resolution offered by them is currently insufficient to resolve mm-cm scale rock breakdown features (Chan et al., 2016).

Stereophotogrammetry is a method of DEM generation using stereo images of an object/surface. It is widely-applied in terrestrial and planetary terrains (Kim and Muller, 2009;Li et al., 2011). The knowledge of camera internal geometry (i.e. sensor type and size), camera calibration parameters and Ground Control Points (GCPs) with known coordinates along with inertial measurement parameters (i.e., yaw, pitch and roll) are critical requirements for stereo photogrammetry to solve collinearity equation and orient photogrammetric model (Taconet and Ciarletti, 2007;Aguilar et al., 2009).

While both methods have been effectively used to analyse rock breakdown at larger scales, both require expensive software (e.g. SocetSet, PHOTOMOD, FARO Scene, Trimble RealWorks, Leica CYCLONE, VisionLidar) and expert knowledge to process data and generate DEMs, the cost of which may push this technology beyond many academic research budgets.

The micro-roughness meter (MRM) (McCarroll, 1992;McCarroll and Nesje, 1996;White et al., 1998) is operated manually and has been used to characterise and quantify breakdown on rock surfaces. Direct physical access to the rock surface is required, which limits sampling in out of reach locations (McCarroll and Nesje, 1996). While the resolution, precision, and accuracy of MRM (~0.001 to 0.005 mm) is higher than laser scanning and photogrammetry techniques (sub-mm to mm), the topographic data obtained from MRM is one dimensional and limits the analysis to the calculation of profile roughness parameters. The profile roughness parameters only provides information along a profile, not entire rock surface which often makes it difficult to determine the exact nature of a topographic feature (Leach, 2013). In comparison, 3D data from laser scanners and photogrammetry enable calculation of areal surface roughness parameters. These parameters have advantages over traditional profile roughness parameters and have more statistical significance than equivalent profile measurements (Leach, 2013).

## 1.1. Structure from Motion (SfM)

Structure from Motion (SfM) is an established and widely used method to generate 3D models in the geosciences (Favalli et al., 2012;Westoby et al., 2012;Smith et al., 2016). It is increasingly used in geomorphology for characterisation of topographic surfaces and analysis of spatial and temporal geomorphic changes, with an accuracy comparable to existing laser scanning and stereo photogrammetry techniques in close range scenario (Aguilar et al., 2009;Thoeni et al., 2014;Smith et al., 2016;Wilkinson et al., 2016). SfM photogrammetry utilises a sequence of overlapping digital images of a static subject taken from different spatial positions to produce a 3D point cloud. Image metadata for image matching is used to estimate 3D geometry and camera positions using bundle adjustment algorithm (Smith et al., 2016). The workflow uses an automated Scale Invariant Feature Transform (SIFT) image matching method (Smith et al., 2016). The advancement in new image matching algorithms has eased and automated the SfM workflow compared to stereophotogrammetry (Remondino et al., 2014;Smith et al., 2016).

Applications in geomorphology include laboratory flume experiments (Morgan et al., 2017), rockslides and landslide (Niethammer et al., 2012;Russell, 2016), eroding badlands (Smith and Vericat, 2015), fluvial morphology (Javernick et al., 2014;Dietrich, 2015;Bakker and Lane, 2016;Dietrich, 2016a, b), peatland

microforms (Mercer and Westbrook, 2016), glacial processe dynamics (Piermattei et al., 2016;Immerzeel et al., 2017), river restoration (Marteau et al., 2016), mapping coral reefs (Casella et al., 2016), beach surveying (Brunier et al., 2016), soil erosion (Snapir et al., 2014;Balaguer-Puig et al., 2017;Prosdocimi et al., 2017;Vinci et al., 2017;Heindel et al., 2018), volcanic terrains (James and Robson, 2012;Bretar et al., 2013;Carr et al., 2018), porosity of river bed material (Seitz et al., 2018), grain size estimation of gravel bed rivers (Pearson et al., 2017)

and coastal erosion (James and Robson, 2012). In addition, SfM has also been widely used in archaeology for photogrammetric recording of small-scale rock art and artefacts, and large-scale archaeological sites (Sapirstein, 2016;Sapirstein and Murray, 2017;Jalandoni et al., 2018;Sapirstein, 2018).

The increased uptake of this method is primarily due to its relatively low cost, high portability, and ease of data processing workflow. Much of the SfM workflow is automated in a range of relatively affordable commercial

software (e.g. Agisoft Photoscan, SURE, Photomodeler), closed source free software (e.g. VisualSfM, CMPMVS), and open source software (e.g., Bundler, OpenMVG, OpenMVS, MicMac, SFMToolkit).

There is a considerable amount of available literature on SfM techniques and workflows. A detailed discussion of the technique is found in several available papers e.g. Westoby et al. (2012); Fonstad et al. (2013); Thoeni et al. (2014); Micheletti et al. (2015a); Micheletti et al. (2015b); Eltner et al. (2016); Ko and Ho (2016); Smith et al.

(2016); Schonberger and Frahm (2016); Bedford (2017); Zhu et al. (2017); Ozyesil et al. (2017).

Several studies have reported high accuracy in 3D topographic data obtained using SfM when compared to methods such as Terrestrial Laser Scanner (TLS) or RTK-GPS surveys (Harwin and Lucieer, 2012;Favalli et al., 2012;Andrews et al., 2013;Fonstad et al., 2013;Nilosek et al., 2014;Caroti et al., 2015;Dietrich, 2015;Palmer et al., 2015;Clapuyt et al., 2016;Koppel, 2016;Piermattei et al., 2016;Panagiotidis et al., 2016;Wilkinson et al.,

2016). A detailed comparison of cost-benefit, data acquisition rate, spatial coverage, operating conditions, resolution and accuracy analysis between TLS and SfM techniques are found in Smith et al. (2016), and Wilkinson et al. (2016). The recent advances in Structure from Motion approaches (SfM) have yet to be been widely applied to micro-scale landforms, such as rock breakdown features.

Here we trial the use of SfM for very high resolution (sub-mm) application. Our approach uses high-resolution digital photography (from consumer grade camera) combined with SfM workflow. We evaluate errors in our DEMs using checkpoints in the field and validate our approach through a series of controlled experiments. We also assess the error propagation with distance from the control target in DEMs generated in our experiment. We find that SfM offers a robust approach for rock breakdown studies.

Our work provides an alternative and/or additional cost-effective, transportable and fieldwork-friendly method for use in geomorphological studies that require the production of high-resolution topographic models from field sites. Below, we outline the development and test of our approach in the field and under controlled conditions. We provide a detailed guide so that others may adopt our approach in their research.


5 **2. Methodology**

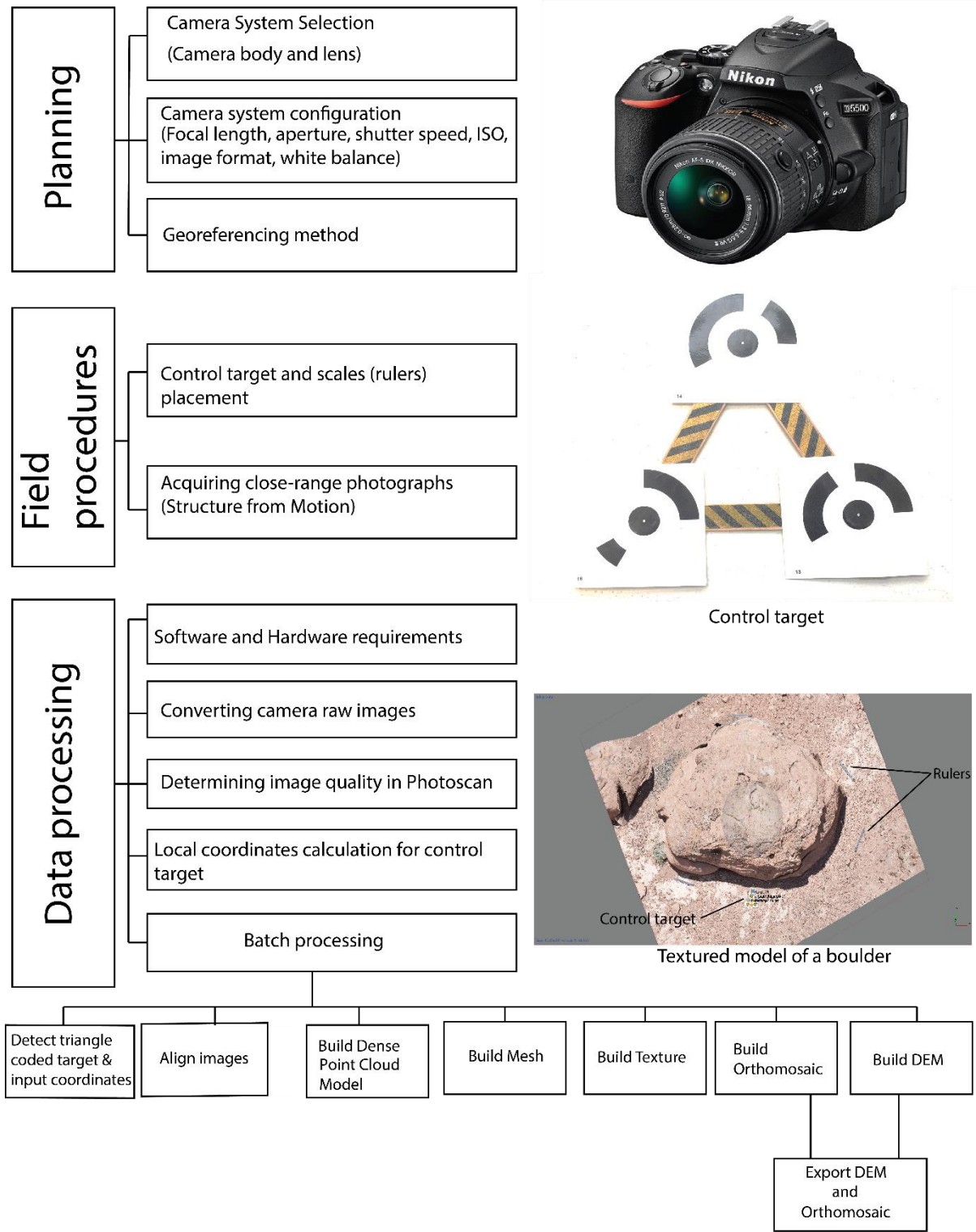

**Figure 1.** A schematic diagram of the typical workflow for Digital Elevation Model (DEM) production described in this study.

## 2.1. Equipment

The quality of image data collection can be improved by careful camera system selection, configuration, and image acquisition. The Camera system plays a vital role in effective resolution, signal-to-noise ratio, and distortion (Mosbrucker et al., 2017). For this work, low-cost, consumer-grade, ultra-compact and lightweight Nikon D5500 DSLR camera was used. A Digital Single Lens Reflex (DSLR) camera system includes a camera body and a lens. This camera has an Advanced Photo System type-C (APS-C) sensor (366.6 mm$^2$) with no anti-aliasing filter and captures an image with an effective resolution of 24.2 Mega Pixels (MP). A DSLR camera provides flexibility in selecting different kinds of lenses and captures high-resolution images in raw (RAW) format. Images in raw format store more Red Green blue (RGB) pixel information than in Joint Photographic Experts Group (JPEG) format. We used a zoom lens with a variable focal length of 18-55 mm and a 35 mm prime or fixed focal-length lens in this study. More comprehensive discussion of camera system consideration and configuration for SfM Photogrammetry work is found in Bedford (2017), Mosbrucker et al. (2017), and Sapirstein and Murray (2017).

## 2.2. Control target and local coordinate system

The dense point cloud generated by SfM is not scaled or oriented to real-world dimension. Therefore, registration to a known coordinate system (geographic or local) using Ground Control Points (GCPs) is required to reference and scale the model. GCP refers to a point with known coordinates (x, y, z). Incorporating GCPs in the SfM workflow is known to reduce systematic errors such as doming and dishing (Javernick et al., 2014;James and Robson, 2014) and permits a check on the accuracy of DEMs. At least three GCPs are required to generate a DEM from a dense point cloud.

For our study, we designed and built a new, portable control target (Figure 2). The triangle control target was made from 13 cm long craft sticks covered with textured plastic tape to protect it from shrinking and swelling in humid conditions (Figure 2). Each vertex served as a GCP. A set of three 12-bit coded markers were printed from Agisoft Photoscan software, laminated and attached at each vertex (Figure 2). The advantage of using coded markers is that they can be automatically identified in Photoscan which minimises the time and reduces error. Goldstein et al., (2015) found that the number and the placement of GCPs affect the accuracy of SfM derived DEMs. In this work, our area of interest was small (<10 m$^2$) hence we determined that three GCPs would be sufficient.

We used our triangle coded control target (GCPs) to calculate local coordinates to scale and reference our DEMs (Figure 2). The length of the triangle sides to the GCP centre was measured using an Engineer's scale with 0.5 mm accuracy. The sides of the triangle were 0.133, 0.132, and 0.131 m respectively for a, b, and c (Figure 2). The angle A (54.03°/1.06 radians) was determined using cosine rule, and the coordinates of each vertex of the triangle were determined using trigonometry (Figure 2 and Table 1).

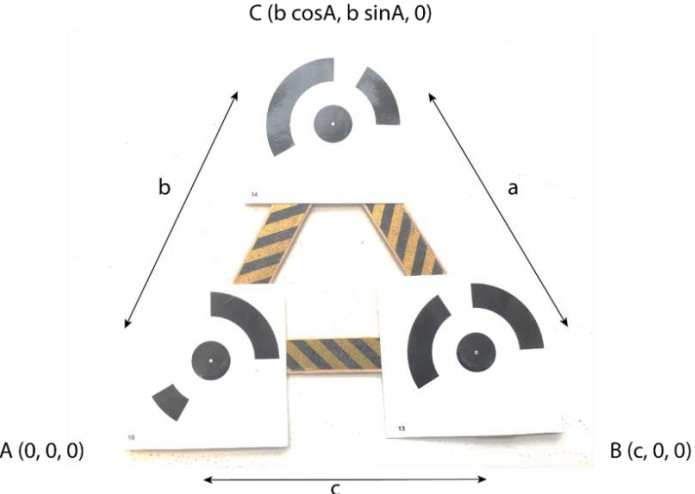

**Figure 2.** Triangle coded control target. Each vertex of this triangle is a GCP.

**Table 1.** Local coordinates for each vertex of triangle coded control target.

| Coded marker | Vertices | X (m) | Y (m) | Z (m) |
|---|---|---|---|---|
| **Target 16** | A position | 0 | 0 | 0 |
| **Target 13** | B position | 0.131 | 0 | 0 |
| **Target 14** | C position | 0.064489 | 0.115175 | 0 |

### 2.3. Data processing

Following image data acquisition (described below) the data were processed using an Intel Xeon workstation with 32 GB of RAM and 2GB Nvidia Quadro 4000 graphics card. We used commercially available software (Adobe Lightroom CC) to process raw images and Agisoft PhotoScan for DEM generation. Photoscan is a 'blackbox' software, so it remains unclear the exact SfM algorithm used.

Photoscan does not support NEF file format (RAW) images generated by the Nikon camera, and they were converted to tiff format. While this step increases processing time, the benefit of capturing images in raw format is that any photometric corrections (i.e., exposure correction) can be performed without losing metadata (Guidi et al., 2014). Raw images were imported into Lightroom and exported as uncompressed tiff image files with AdobeRGB (1998) colour space (Süsstrunk et al., 1999;Korytkowski and Olejnik-Krugly, 2017) and 16 bits/component bit depth. Image histograms generated in Lightroom confirmed that the images were well exposed, and no photometric correction was required. Each RAW file was 25-30 MB. When converted to uncompressed tiff, this increase to 130-140 MB per image file. Exporting tiff images from Lightroom took about 5-10 minutes in total.

### 2.4. DEM generation workflow in Photoscan

Agisoft Photoscan is a popular software for generating DEMs from SfM photogrammetry technique. Many published studies have already described DEM production workflow in Photoscan (e.g. Leon et al. (2015); James et al. (2017a)) so we only summarise the parameters used in our study here. A detailed step by step guideline for this study is presented in Section S1 (supplement). For a more detailed explanation of workflow in Photoscan, we

 refer readers to Agisoft (2016) and Shervais K. (2016).

**Table 2.** Summary of processing parameters in the development of DEM in Photoscan

| | | |
|---|---|---|
| | **General** | |
| | Images | Loading images, image quality determination, images with quality index <0.5 discarded |
| **Stage 1** | Identification of markers: scale bar and coordinate input | Coded markers detected, local coordinates entered, scale bar created |
| | Measurement and scale bar accuracy setting | Measurement and scale bar accuracy adjustment, 0.01 mm for experiments, 0.5 mm for field data |
| | Masking | Only if images contain unwanted scenes |
| | Coordinate system | Local Coordinates (m) |
| | **Alignment parameters** | |
| | Accuracy | Highest |
| | Pair preselection | Generic |
| **Stage 2** | Key point limit | 40,000 |
| | Tie point limit | 4,000 |
| | Constrain features by mask | No (yes if images were masked) |
| | **Optimization parameters** | |
| | Parameters | $f$, $cx$, $cy$, $k1$-$k3$, $p1$, $p2$ |
| | **Dense point cloud Reconstruction parameters** | |
| **Stage 3** | Quality | High |
| | Depth filtering | Mild |
| | **Mesh Model Reconstruction parameters** | |
| | Surface type | Height field |
| | Source data | Dense |
| | Interpolation | Enabled |
| | Quality | High |
| | Depth filtering | Mild |
| | Face count | 11,536,078 |
| | **Texturing parameters** | |
| | Mapping mode | Generic |
| **Stage 4** | Blending mode | Mosaic |
| | Texture size | 4,096 x 4,096 |
| | **DEM Reconstruction parameters** | |
| | Coordinate system | Local Coordinates (m) |
| | Source data | Dense cloud |
| | Interpolation | Enabled |
| | **Orthomosaic Reconstruction parameters** | |
| | Coordinate system | Local Coordinates (m) |
| | Channels | 3, uint16 |
| | Blending mode | Mosaic |
| | Surface | Mesh |
| | Enable color correction | Yes |

## 3. Error evaluation experiments

A series of controlled image acquisition experiments were conducted to evaluate the horizontal and vertical errors of the DEMs generated using the GCP developed in this study (Figures 3 and 4). In addition, we tested the influence of a range of other variables on the accuracy and quality of DEMs. These include: (1) Type of lens, (2) Prior lens profile correction, (3) colour space of images, (4) dense point cloud quality setting in Photoscan, (5) image file format, (6) the position of control target with respect to subject, and (7) masking of images (Table 3).

### 3.1. Experiment design

In order to validate the sub-mm horizontal and vertical accuracy of DEM generated, a calibrated error evaluation chart was designed in Adobe Indesign and printed as $1.4 \times 1.4$ m poster (Figure 3). This chart contains four concentric squares and 16 coded scale bars of known length (Figure 3). This chart was laid on the relatively flat ground ($\pm 1°$ from the centre of the poster), and 16 wooden cubes of dimension ~5 cm were placed on vertices of each square (Figure 3 and 5a). We chose wooden blocks because of their non-homogeneous texture which is easily reconstructed using photogrammetry. Sixteen coded scale bars were used as checkpoints to estimate horizontal (XY) errors, and sixteen wooden blocks were used to determine vertical (Z) errors. Two triangle coded control targets were designed in the centre and the left corner of the poster (Figure 3). An additional four, 25 cm long coded scale bars were placed 60 cm away from the outer scale bar on each side of the poster. The coordinates of the triangle coded target were determined as described in section 2.2. The experiment was undertaken outside in overcast lighting conditions.

Three sets of images of the poster and nearby ground surface made up of concrete paving stones with visible edges were acquired using a zoom lens set at 24 mm and a 35 mm prime or fixed focal-length lens. Two sets of images were taken by zoom lens set at 24 mm and 35 mm prime lens. The third set of images were acquired, using zoom lens set at 24 mm, to cover the extended area where four additional scale bars were placed outside the poster on the cement surface. All the images were acquired using Nikon D5500 in manual mode. Camera settings were adjusted for the best result for the lighting conditions during the experiment. Aperture was set at f/7.1, shutter speed was fixed at 1/200 s, and ISO was kept at 100. The focus was set to auto-focus during image acquisition. Images were acquired in raw and then converted into an uncompressed tiff in Adobe Lightroom (section 2.3).

RAW images were processed to change a few parameters in the image sets. Ten models were run in Photoscan from the three sets of images acquired. The DEMs were generated using the workflow described in section 2.4. Table 3 summarises the variables tested in the ten DEMs.

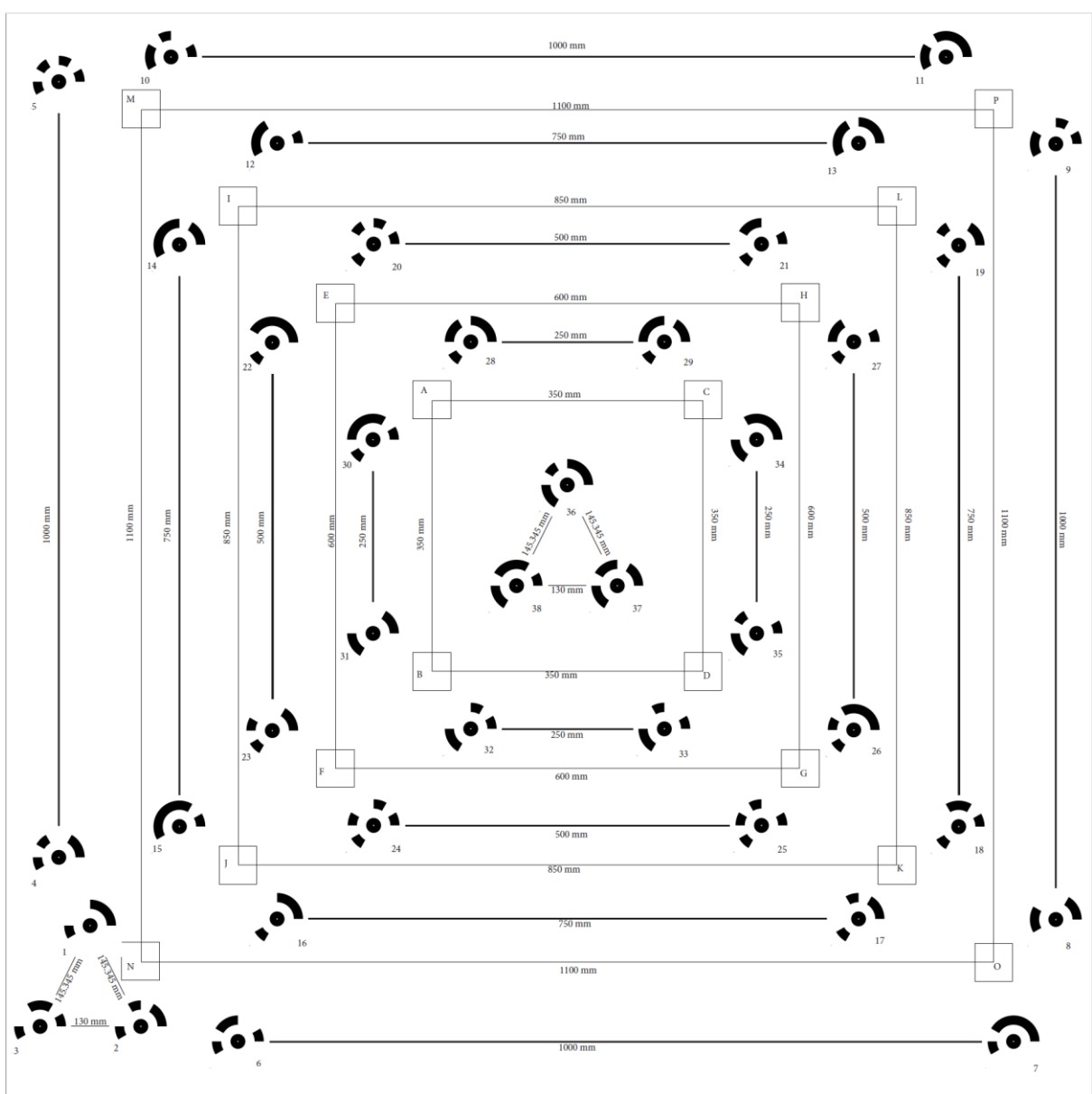

**Figure 3.** Error evaluation chart (1.4 × 1.4 m). Coded scale bars are horizontal checkpoints. Wooden blocks (small squares) at each vertex of bigger squares are vertical checkpoints. Two triangle coded control targets were used to georeferenced DEMs.

**Table 3.** Experimental design used in error evaluation experiment. Cross mark represents the DEMs compared for a variable.

| Variables tested → | Type of lens (zoom lens vs prime lens) | Prior lens profile correction in images | Colour space (e.g. ProPhotoRGB, sRGB, AdobeRGB) | Dense point cloud quality setting (e.g.ultra high,high, medium) | Image format (tiff vs jpeg) | Position of the control target | Masking of images |
|---|---|---|---|---|---|---|---|
| **DEMs ↓** | | | | | | | |
| 24 mm extended area | | | | | | | |
| 24 mm profile corrected | | × | | | | | |
| 24 mm without profile corrected | × | × | | | | | |
| 35 mm AdobeRGB | | | × | | | | |
| 35 mm sRGB | | | × | | | | |
| 35 mm ProPhotoRGB | | × | × | × | × | | |
| 35 mm jpg | | | | | × | | |
| 35 mm profile corrected | | × | | | | | |
| 35 mm masked | × | | | × | | × | × |
| 35 mm corner control target | | | | | | × | × |

### 3.2. Estimating errors

The error evaluation chart (Figure 3) was used to estimate errors in the following way. The coded scale bars were used as horizontal checkpoints. The distance between coded markers and the centroid of the triangular control target was measured with an accuracy of 0.01 mm in Adobe InDesign. The scale bars were automatically detected in Photoscan. These scale bars were not used to scale or optimise the sparse point cloud in Photoscan. Photoscan estimated the length of coded scale bars based on the referencing information from the control target. The known

length of coded scale bars was subtracted from the estimated length in Photoscan to calculate the horizontal error.

To determine the vertical error of the DEMs, the wooden blocks were used as checkpoints (Figure 5a). The DEMs and orthophotos were imported in ArcMap 10.4.1 (Figure 5). The height of wooden blocks was measured in ArcMap using the Interpolate Line tool (3D Analyst tool), by drawing a straight line across one of the sides of the wooden block and extending it to the ground surface. Height was estimated as the difference in mean elevation

between wooden block top surface and the surrounding ground surface on each side. The actual height of wooden blocks was measured by an electronic digital Vernier Caliper. The Vernier Caliper has an accuracy of 0.03 mm and measurement repeatability of 0.01 mm. We obtained five measurements along the same side of wooden block measured in ArcMap. We take the mean of these five measurements to calculate the height of the wooden block.

The measured height was subtracted from the estimated DEM height to calculate the vertical error. The distance between the centre of wooden blocks and centroid of the triangle coded target was determined in Adobe Indesign. We used horizontal and vertical checkpoint errors with their distance from the control target to visualise error propagation in DEMs with distance (section 3.3.1).

### 3.3. Experiment Results

### 3.3.1. Distribution of horizontal and vertical errors

Error propagation with distance was estimated, and the data are shown in Tables S1-S2. The horizontal checkpoint errors for 24 mm extended area and 35 mm masked DEMs (Table S1) were used to visualise errors over an area of 6.14 $m^2$ and 1.96 $m^2$ respectively as a contour plot (Figure 4 a and c). The data show that horizontal errors are

almost symmetrical in X and Y direction (Figure 4 a and c). We used vertical checkpoints for 24 mm extended area and 35 mm masked DEMs (Table S2) to visualise vertical errors as the surface plot over an area of 1.96 $m^2$ (Figure 4 b and d).

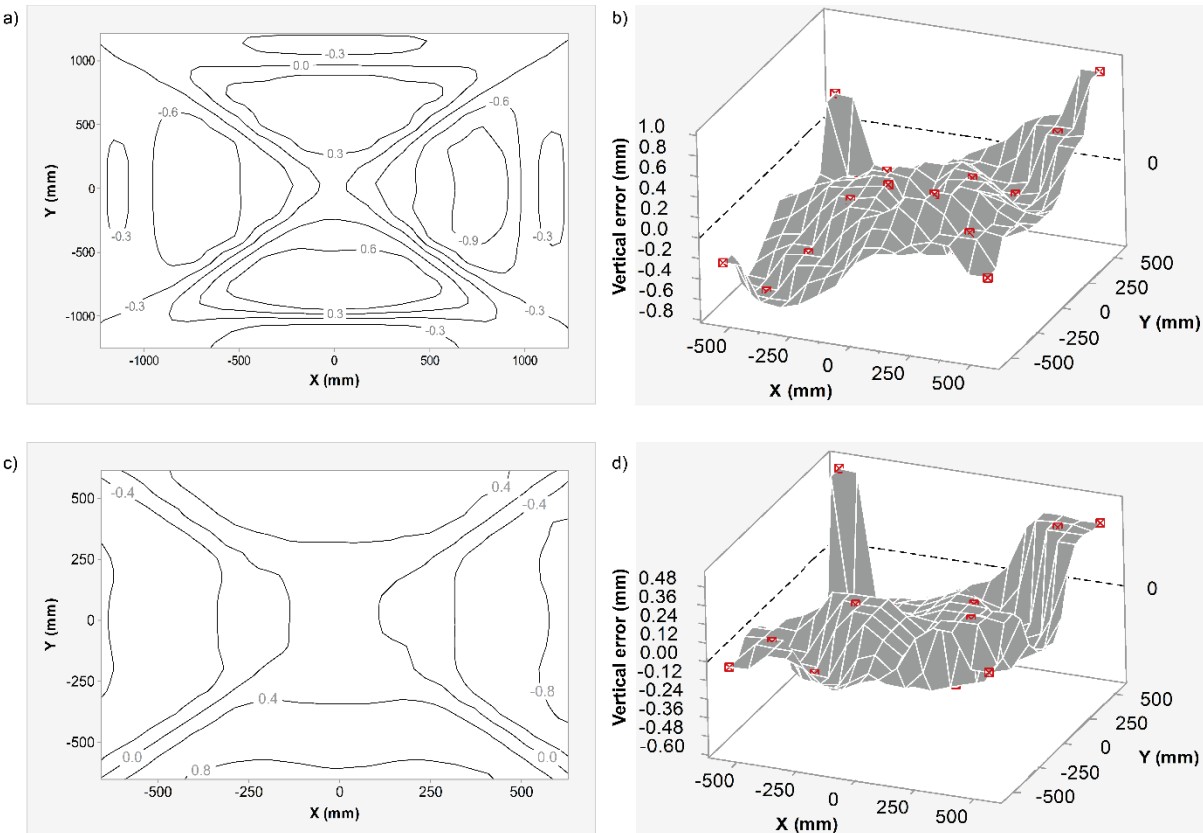

**Figure 4.** (a) Horizontal error contour plot for a DEM generated using images acquired with the zoom lens.
Contours represent horizontal (XY) error (in mm) in the DEM. (b) Vertical errors in DEM generated using images from a zoom lens. Red cubes on the surface in the plot shows the location of wooden blocks (vertical checkpoints). (c) Horizontal error contour plot for a DEM generated using images from a prime lens. Contours represent horizontal (XY) error (in mm) in the DEM. (d) Vertical errors in DEM generated using images from a prime lens. Red cubes on the surface in the plot shows the location of wooden blocks (vertical checkpoints).

### 3.3.2. Role of image variables in DEM error

In this section, we present the findings from our DEM error evaluation experiment. Orthophoto and DEM of the error evaluation chart are shown in Figure 5. The summary of ten DEMs produced in the error evaluation experiment is presented in Table 4.

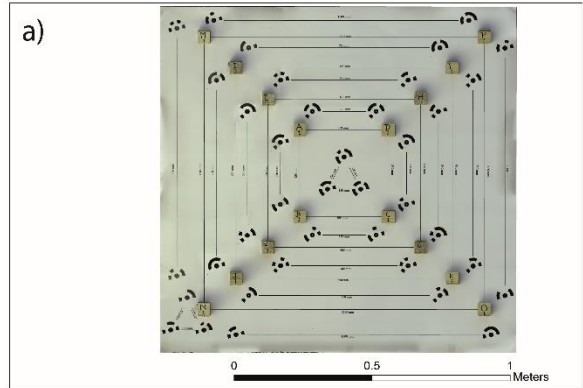
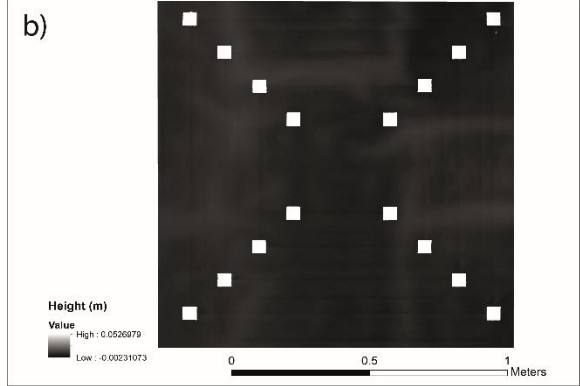

**Figure 5.** (a) Orthophoto of error evaluation chart. Coded scale bars represent horizontal checkpoints. Wooden blocks denote vertical checkpoints. (b) DEM of the error evaluation chart.

**Table 4.** Summary Experimental DEM data. Reprojection Error is the root mean square reprojection error (RMSE) averaged over all tie points on all images. In some publications, reprojection error is also referred to as RMS image residual (James et al., 2017a). In Agisoft manual, reprojection error is defined as the distance between the point on the image where a reconstructed 3D point can be projected, and the original projection of that 3D point detected on the photo (Agisoft, 2016). It is used to quantify how closely an estimate of a 3D point recreates the point's original projection and as a basis for the 3D point reconstruction procedure. Root Mean Square Error (RMSE) XY is the root mean square error for X and Y coordinates for control location/checkpoint. Root Mean Square Error (RMSE) Z is the root mean square error for X and Y coordinates for control location/checkpoint. Projection Accuracy (in pixels) is the root mean square error for X, Y coordinates on an image for control location/checkpoint averaged over all the images.

| DEM | Colour space | No. of images | RMSE X,Y (Check) (mm) | RMSE Z (Check) (mm) | RMSE X,Y (Control) (mm) | RMSE Z (Control) (mm) | Reprojection error (pix) | Projection accuracy (control) (pix) | Accuracy (check) (pix) | Resolution (mm/pix) | Point density (pts/mm²) | DEM quality setting | Time taken (Hr) |
|---|---|---|---|---|---|---|---|---|---|---|---|---|---|
| 24 mm extended area | ProPhotoRGB | 259 | 0.52 | 0.35 | 0.08 | 0.17 | 0.81 | 0.28 | 0.37 | 0.51 | 3780 | High | 169 |
| 24 mm profile corrected | ProPhotoRGB | 178 | 0.56 | 0.37 | 0.08 | 0.16 | 0.72 | 0.29 | 0.35 | 0.45 | 4810 | High | 59 |
| 24 mm without profile corrected | ProPhotoRGB | 178 | 0.55 | 0.35 | 0.08 | 0.17 | 0.77 | 0.29 | 0.37 | 0.45 | 4910 | High | 62 |
| 35 mm AdobeRGB | AdobeRGB | 236 | 0.59 | 0.3 | 0.08 | 0.09 | 0.53 | 0.15 | 0.19 | 0.65 | 2330 | Medium | 10 |
| 35 mm corner coded target | ProPhotoRGB | 236 | 0.59 | 0.27 | 0.02 | 0.54 | 0.54 | 0.17 | 0.2 | 0.32 | 9230 | High | 67 |
| 35 mm jpg | ProPhotoRGB | 236 | 0.91 | 0.34 | 1.16 | 3.18 | 3.19 | 0.75 | 1.72 | 0.67 | 2180 | Medium | 8 |
| 35 mm masked | ProPhotoRGB | 236 | 0.59 | 0.28 | 0.08 | 0.08 | 0.62 | 0.16 | 0.19 | 0.32 | 9760 | High | 67 |
| 35 mm profile corrected | ProPhotoRGB | 236 | 0.59 | 0.39 | 0.08 | 0.07 | 0.54 | 0.17 | 0.2 | 0.66 | 2290 | Medium | 9 |
| 35 mm ProPhotoRGB | ProPhotoRGB | 236 | 0.59 | 0.41 | 0.08 | 0.06 | 0.53 | 0.16 | 0.2 | 0.65 | 2330 | Medium | 10 |
| 35 mm sRGB | sRGB | 236 | 0.59 | 0.42 | 0.08 | 0.62 | 0.54 | 0.16 | 0.19 | 0.65 | 2330 | Medium | 10 |

Below we also explored the role of several parameters on the accuracy of DEMs and the detailed results are in Section S2 (supplement). Horizontal and vertical checkpoint errors are used to compare these DEMs.

Although, our experiment suggests that there is no statistically significant difference in the accuracy of DEMs generated from prime and zoom lens we find that the use of the prime lens will yield lower errors compared to a zoom lens for SfM photogrammetry. Our results also indicate that prior lens profile correction, placement of
control target relative to the subject of interest and masking of images had no statistically significant effect on the accuracy of DEM. However, we report that using Adobe RGB colour space and tiff file compression reduced error in DEMs (Table 4). We obtained better resolution and accuracy using "High" dense point cloud quality setting in Photoscan. Based on our findings, we use these parameters in our field survey.

### 3.3.3.    Repeatability

We used two independent image surveys to test the repeatability of our DEM generation method. We obtained a very high intraclass correlation for horizontal (ICC=0.999), and vertical (ICC=0.911) checkpoint errors between two DEMs produced from two different set of images (24 mm extended area and 24 mm without profile corrected). These DEMs were generated using identical image parameters and settings in Photoscan. Therefore, this method
of DEM generation can easily be repeated.

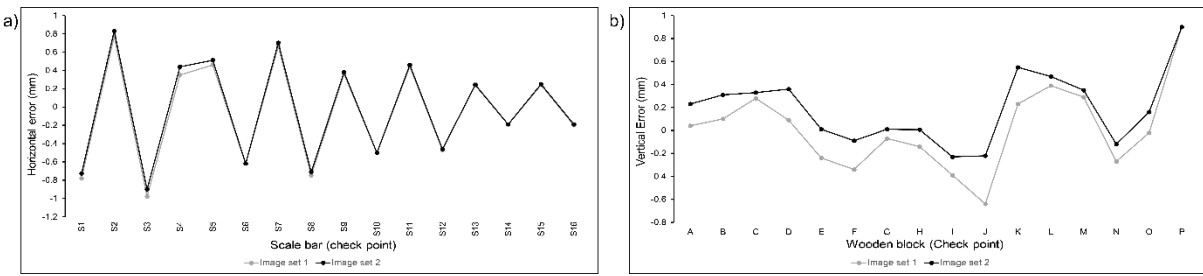

**Figure 6.** a) and b) Horizontal checkpoint and vertical checkpoint errors in DEMs produced from two different image sets to test repeatability.

Additionally, we performed DEM of Difference (DoD) on these two DEMs (24 mm extended area - 24 mm
without profile corrected) of the same subject generated from two independent image surveys. The change in vertical elevation for the evaluation chart and surrounding ground surface made up of concrete paving stones was calculated from the DoD, (Fig 7). The change in elevation (E) is within Limit of Detection (LoD) and is interpreted as no change (±0.49 mm) and the change above the LoD value is interpreted as change (-0.49>E>0.49 mm). We find that the nearby textured concrete ground surface which had good number of keypoints during sparse point
cloud generation shows no change. The shadow areas within the sides of wooden blocks, the edges of wooden blocks and flat and textureless evaluation chart area that had poor image match and thus low keypoints shows changes. Cullen et al. (2018) have demonstrated that the reliability of SfM to detect sub-mm changes depends on texture and complexity of the rock surface. SfM is known to less reliable in reconstructing non-textured, reflective and flat objects or scenes (Agisoft, 2016). We notice that these changes are not related to the distance from the
control target but areas with poor image matching due to homogeneous texture and shadows. The rock surface

5    with non-homogeneous texture will produce better image matches and thus improve model quality and accuracy. The result show that with our control target approach it is possible to generate DEM with sub-mm accuracy, but this will depend on the complexity and texture of the surface. Using our approach Cullen et al. (2018) successfully generated DEMs of simulated rock surface (~100 cm$^2$) with sub-mm accuracy.

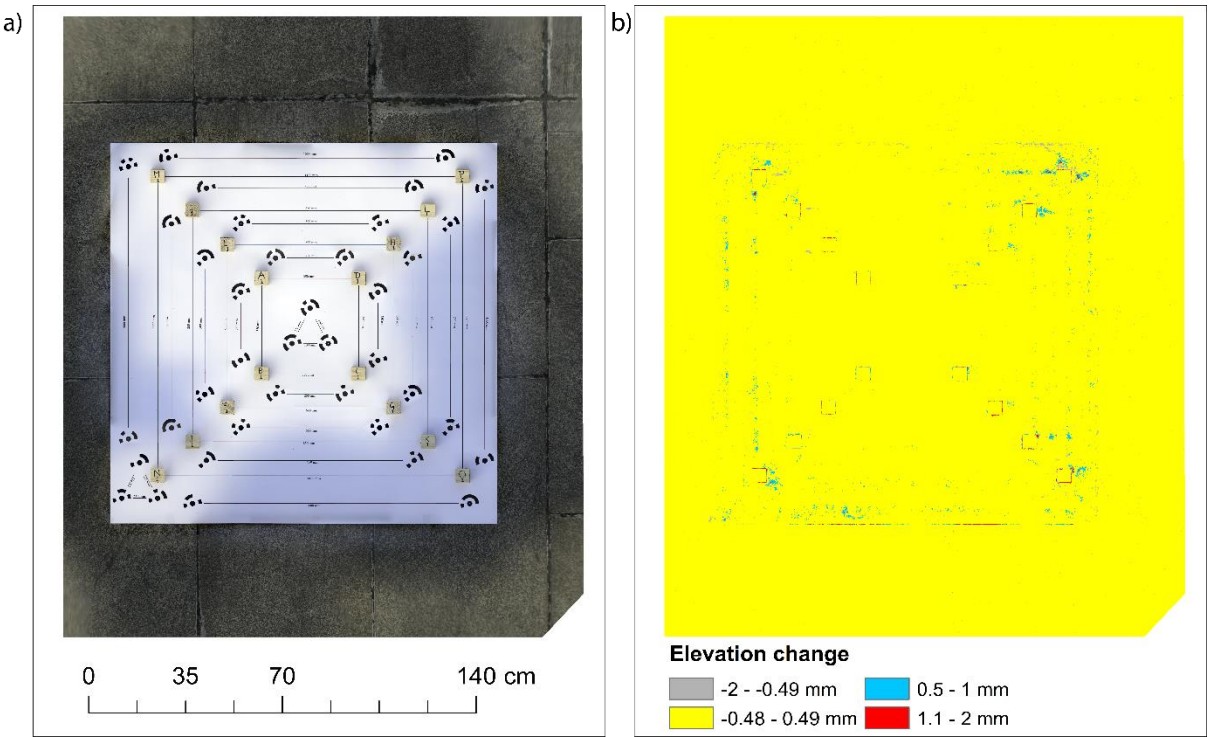

10   **Figure 7.** (a) The orthophoto showing evaluation test chart and nearby ground surface area. (b) DoD showing a change in surface elevation between two independent DEMs. The yellow coloured area is within the LoD (±0.49 mm) and is interpreted as no change.

### 4.   Field application of SfM for DEM generation

We tested the approach on eight Moenkopi Sandstone outcrops (intermediate axis = ~ 2 m) at a field site near
15   Meteor Crater, Arizona. Meteor Crater is located in a relatively low-relief, southern part of the Colorado Plateau near the town of Winslow in north-central Arizona (35° 1' N, 111° 1' W) (Shoemaker and Kieffer, 1979;Shoemaker, 1987). Moenkopi is very fine-grained reddish-brown sandstone (Kring, 2017). These outcrops have weathered to produce surfaces with different shapes, sizes, aspect, slope and contain a range of weathering features such as pits, alveoli, flaking, crumbling, fractures, colouration, and lichen colonisation (see Chapter 6).

## 4.1. Data collection

We used the zoom lens set at 24 mm focal length (36 mm full frame camera equivalent). The focal length of 24 mm was chosen as it provided a greater field of view where there was little space to move around to take images in the field (e.g., very steep slope). Camera aperture was set to f/6.3. A smaller aperture allows less light to reach the camera sensor and gives a larger depth of field (Haukebø, 2015). An image with larger depth of field is sharper and has a larger area in focus and are recommended for photogrammetry work (Bedford, 2017). A higher shutter speed (1/400) was chosen to compensate camera shake due to, e.g., the wind. ISO was kept at 100 to minimise noise in the images (Mosbrucker et al., 2017). White balance was kept at daylight mode. During photo acquisition, care was taken to ensure that image was sharp and everything in the frame was in focus. Matrix metering mode was selected to provide the best exposure and equal brightness throughout the image. Images were taken in autofocus mode to maintain optimal image quality (sharpness). These settings were chosen based on the lighting and field conditions, and field testing demonstrated high image quality at these settings.

Several images were acquired from different vantage points. Firstly, from all around the boulder surface (from a distance of ~2 m) followed by additional close-range (from a distance of <~1 m) images (see Figure 8). Images were acquired with at least 60% lateral overlap. The theoretical minimum number of images required in SfM workflow is 3 (Favalli et al., 2012;Westoby et al., 2012). However, there is no maximum limit to the number of input images in the SfM workflow. The number of images required to reconstruct accurate dense point cloud depends on the size and complexity (e.g. shape, surface texture, curvature, and slope) of the outcrop. It is always better to take more images as it will permit less sharp images to be discarded before processing.

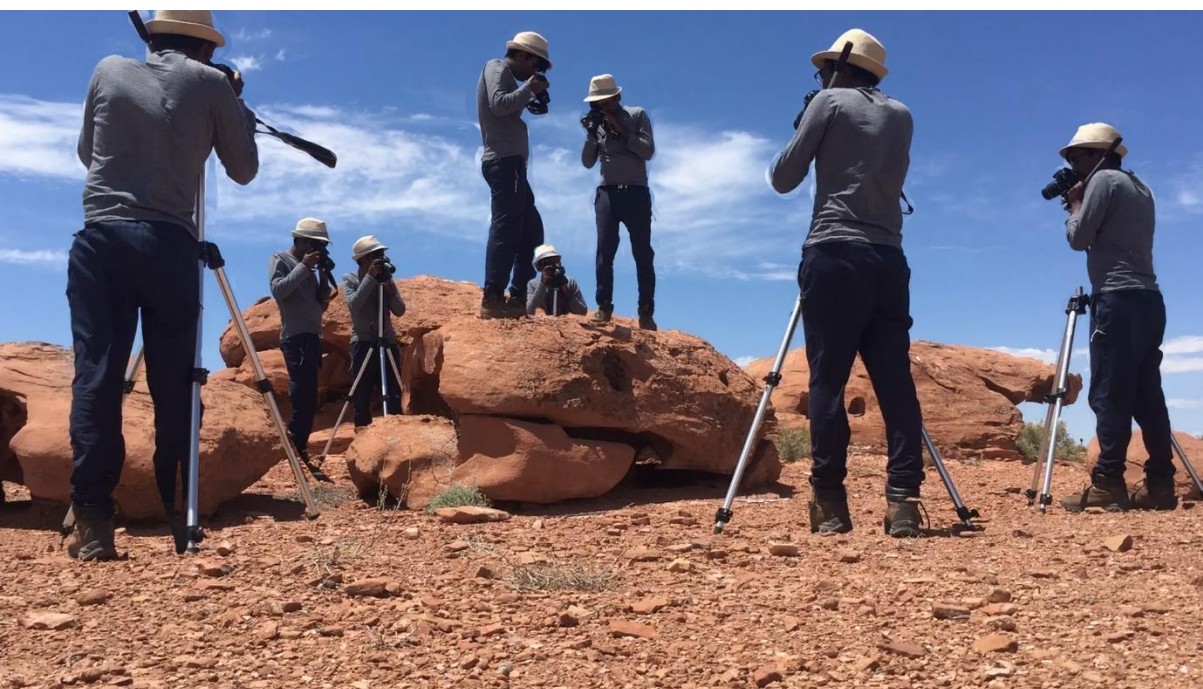

**Figure 8.** Multi-exposure image showing the different spatial positions from where images were acquired. First images were acquired from a distance of ~2 m then close-up images were taken from a distance of <~1 m. DEM and orthophoto of the imaged boulder is shown in Figure 10 g and h.

For a detailed guideline for ideal image acquisition in the field we recommend the following: (Smith et al.,
2016;Bedford, 2017;Mosbrucker et al., 2017). For our data collection, a triangle coded control target (Figure 2)
was placed on the ground parallel to the top surface of the boulder (Figure 1). It is crucial for the control target to
be flat and approximately parallel to the surface of interest as it defines the orientation of the surface of interest in
the DEM. If the adjacent ground is not level, the control target can be placed on top of the target surface. We used
four rulers of 30 cm and placed them around the outcrops (Figure 1). These rulers were used as checkpoints to
estimate horizontal errors in the DEMs. The images were acquired in quick succession in the field to ensure that
there was a minimum change in the shadow lengths and lighting conditions. We acquired images during early
morning and evening and tried to avoid shadows in the image. The images were shot in raw format. A potential
limitation to this in the field is that they take up to twice as much storage space as JPEGs. For an area ~10m$^2$,
placement of GCPs, rulers and image acquisition took approximately 20 minutes. Images were processed as
described in section 2.4. DEM and Orthophoto generation took 8-10 hours on "high" dense point cloud quality
setting.

### 4.2. Field Results

### 4.2.1.    DEMs of Moenkopi outcrops in the field

We generated eight DEMs and orthophotos of weathered Moenkopi outcrop surfaces (Figure 9 and 10). We find
that small weathering features, such as weathering pits (mm scale), are clearly resolved in our DEMs and
Orthophotos (Figure 9 and 10).  Details of DEM parameters have been summarized in Table 5. Horizontal errors
for checkpoints were calculated by measuring the length of rulers from orthophoto in Photoscan and subtracting
the known length of the ruler from it. The distance of the checkpoints from control target was measured in
Photoscan.  Horizontal error propagation with distance from the control target in DEMs is presented in Table 6.
The resolution of DEMs ranges from 0.45 to 0.68 mm/pixel. All the Orthophotos have a resolution of 0.5
mm/pixel. Horizontal and vertical RMSE of control points is less than 0.5 mm except for vertical error for boulder
S2-M2 (Table 5). Horizontal RMSE estimated from checkpoints were also less than 0.5 mm (Table 5).

5    **Table 5.** Field DEM data summary

| DEM | Boulder dimension (m) | No. of Images | Resolution (mm/pix) | Reprojection Error (pix) | RMSE XY (control) (mm) | RMSE Z (control) (mm) | Projection accuracy (control) (pix) | RMSE XY (check points) (mm) |
|---|---|---|---|---|---|---|---|---|
| S2-M1 | 2.64 × 1.91 | 62 | 0.68 | 0.45 | 0.26 | 0.28 | 0.23 | 0.31 |
| S2-M2 | 1 × 0.9 | 47 | 0.49 | 0.56 | 0.28 | 0.88 | 0.29 | 0.22 |
| S2-M3 | 1.82 × 1.23 | 55 | 0.61 | 0.46 | 0.28 | 0.14 | 0.29 | 0.21 |
| S2-M4 | 1.31 × 1.04 | 48 | 0.45 | 0.50 | 0.31 | 0.64 | 0.14 | 0.15 |
| S2-M5 | 1.38 × 1.05 | 55 | 0.55 | 0.67 | 0.26 | 0.1 | 0.29 | 0.19 |
| S2-M7 | 2.5 × 1.6 | 59 | 0.51 | 0.48 | 0.35 | 0.39 | 0.21 | 0.13 |
| S2-M20 | 3.2 × 1.8 | 52 | 0.60 | 0.41 | 0.32 | 0.28 | 0.09 | 0.25 |
| S3-M33 | 5 × 2.9 | 66 | 0.58 | 0.30 | 0.32 | 0.38 | 0.06 | 0.27 |

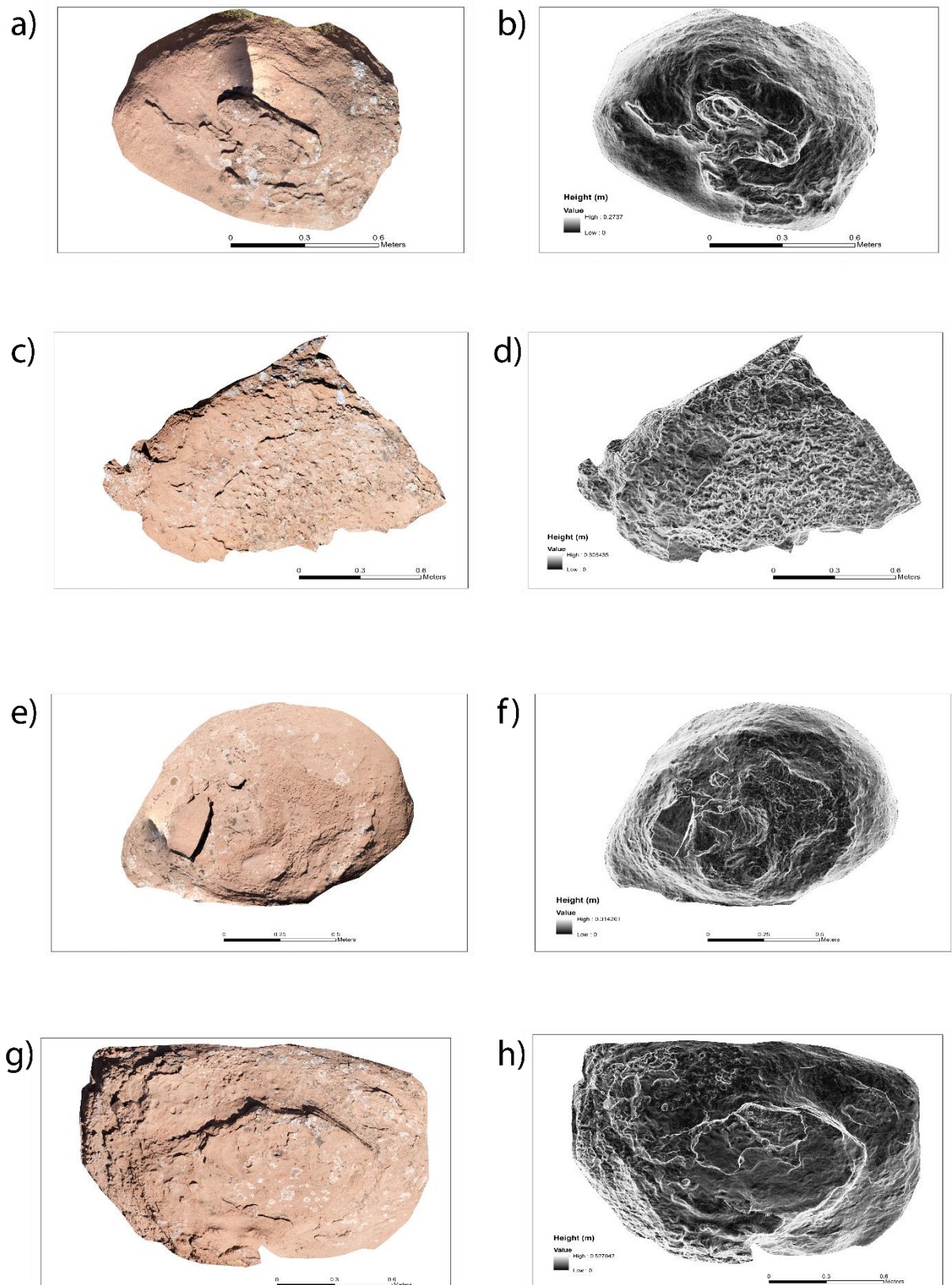

**Figure 9.** Orthophotos and DEMs of Moenkopi outcrops. (a) and (b) Boulder S2-M2. (c) and (d) Boulder S2-M5. (e) and (f) Boulder S2-M4. (g) and (h) Boulder S2-M3.

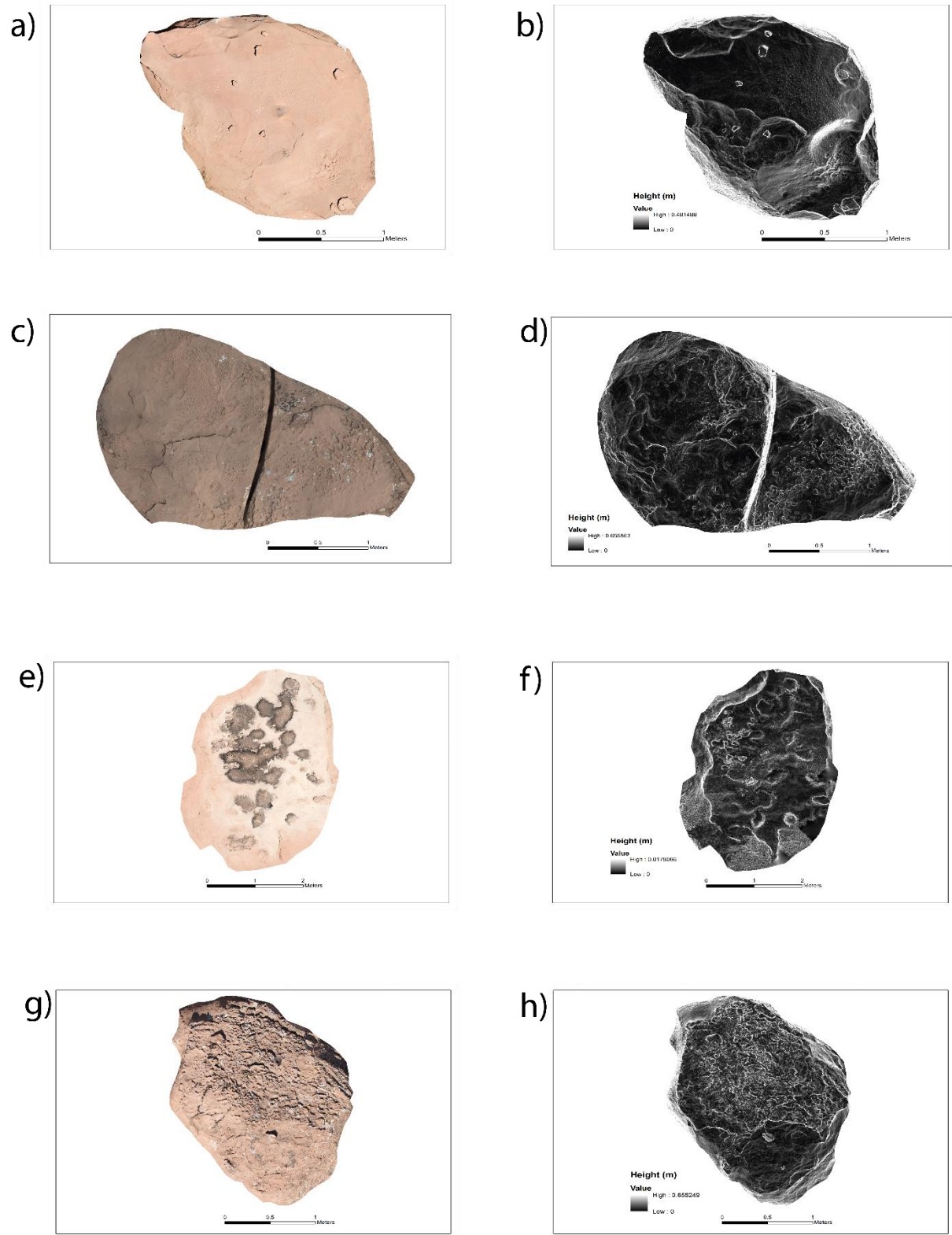

**Figure 10.** Orthophotos and DEMs of Moenkopi outcrops. (a) and (b) Bedrock S2-M7. (c) and (d) Bedrock S2-M20. (e) and (f) Bedrock S3-M33. (g) and (h) Boulder S2-M1.

5 **Table 6.** Horizontal error propagation with distance from control target in DEMs from the field.

| Distance from the target (m) | S2-M1 (mm) | Distance from the target (m) | S2-M2 (mm) | Distance from the target (m) | S2-M3 (mm) |
|---|---|---|---|---|---|
| 1.39 | 0.1 | 0.57 | 0.2 | 1.59 | 0.2 |
| 0.8 | 0.1 | 0.63 | 0.2 | 0.31 | 0.2 |
| 2.5 | 0.1 | 1 | 0.3 | 1.42 | -0.1 |
| 3.4 | -0.6 | 1.2 | 0.2 | 1.95 | 0.3 |

| Distance from the target (m) | S2-M4 (mm) | Distance from the target (m) | S2-M5 (mm) | Distance from the target (m) | S2-M7 (mm) |
|---|---|---|---|---|---|
| 1.1 | 0.1 | 1.1 | -0.2 | 2.47 | 0.1 |
| 0.71 | 0.1 | 1.18 | -0.1 | 1.9 | -0.1 |
| 1.24 | 0.2 | 1.55 | 0.1 | 0.56 | 0.2 |
| 1.43 | 0.2 | 1.41 | -0.3 | 1.95 | 0.1 |

| Distance from the target (m) | S2-M20 (mm) | Distance from the target (m) | S2-M33 (mm) |
|---|---|---|---|
| 0.45 | 0.2 | 5.66 | -0.1 |
| 1.48 | 0.3 | 4.21 | 0.4 |
| 2.48 | 0.3 | 2.17 | 0.2 |
| 2.68 | -0.2 | 2.72 | 0.3 |

## 5. Discussion

There are significant technical and logistical challenges that have resulted in geomorphologists not directly capturing the topographic data of outcrops at the microscale (mm) in the field (Ehlmann et al., 2008). In order to generate high-resolution DEMs (~mm accuracy) of the relatively small boulder and bedrock surfaces (areas < 10 m$^2$), geographic coordinates cannot be used to register SfM dense point cloud. GPS surveying is used to collect topographic point data from surfaces which can be used to register SfM point dense point cloud to build a DEM. The surveying equipment can be expensive (e.g. dGPS, RTK-GPS, and total station). These survey instruments (except total station) have centimetre accuracy which is inadequate for generating DEMs of sub-mm accuracy. The equipment can be challenging to transport in poorly accessible field terrains and can rely on satellite signals which may not work in all locations or global locations. In addition, the equipment requires a relatively low gradient, stable surface to set up. A relatively new approach known as 'direct georeferencing' only requires the camera orientation parameters and GPS (Carbonneau and Dietrich, 2016). However, it can only provide centimetre accuracy which is coarser than needed for small-scale weathering feature analysis.

Total station can be used to determine the coordinates of an unknown point relative to a known coordinate if a direct line of sight can be established between the two points. Coordinates obtained from a total station can be used to register SfM dense point cloud to generate high-resolution DEMs (mm accuracy). However, operating a total station in challenging field conditions have drawbacks similar to those of dGPS survey equipment described above.

A number of previous studies have produced mm-cm resolution DEMs with mm-cm horizontal and vertical accuracy (Favalli et al., 2012;James and Robson, 2012;Bretar et al., 2013;Snapir et al., 2014;Haukebø, 2015;Leon et al., 2015;Micheletti et al., 2015a;Balaguer-Puig et al., 2017;Prosdocimi et al., 2017;Vinci et al., 2017;Smith and Warburton, 2018). These studies employed relatively complicated methods for georeferencing SfM dense point cloud to generate DEM, for example, Favalli et al. (2012) and Micheletti et al. (2015a) used laser scanner coordinate system, Snapir et al. (2014) used laser range finder and optical level to find the relative 3D positions of the GCPs, Bretar et al. (2013) employed stereo-photogrammetric method using a measuring tape for scaling the model, Haukebø (2015) measured 3D positions of each camera positions which are difficult to replicate in the field, James and Robson (2012) utilised distance measured between multiple points on a turntable to scale the model of a sample of size 10 cm in the lab, Prosdocimi et al. (2017) used RTK-GPS to reference DEMs of soil plots (0.25 m$^2$) in the field, Leon et al. (2015) used handheld GPS and several GCPs to scale the DEMs, and others (Balaguer-Puig et al., 2017;Vinci et al., 2017;Smith and Warburton, 2018) used a total station to estimate coordinates for GCPs.

The method presented by Snapir et al. (2014) is useful for making DEMs of a horizontal surface, but difficult to replicate on remote and treacherous field terrain (e.g. slope of mountain, crater or canyon wall). This is due to the difficulty of placing several GCPs and determining their relative position with sub-mm accuracy in these terrains. Another problem of using many GCPs for smaller surface (<5 m$^2$) is that it may cover the area of interest and obscure the DEM of the target surface for further analysis. In this study, we have solved this problem by using a small triangle control target (area ~75 cm$^2$, Figure 2) to georeference the dense point cloud used to generate DEMs with high accuracy. In our experience, we found that using three arbitrary points separated by a longer distance

 (few metres) and using these points to find relative coordinates with each other can be difficult in the field due to curvature and slope of the rock surfaces (e.g. Heindel et al. (2018)). In comparison, this study achieved sub-mm horizontal (<0.5 mm) and vertical (<1 mm) accuracy in sub-mm resolution DEM using a relatively simple georeferencing approach (section 2.2) without any expensive and bulky survey equipment. The DEMs generated following our methodology have sufficient resolution for measurement and quantification of mm-cm scale rock

 breakdown features.

### 5.1. High-resolution DEMs with low errors

Scaling errors in DEMs are important as they will affect any 2D distance or 3D volume measurements obtained from the DEMs (Carbonneau and Dietrich, 2016). Uncertainties in the DEMs are linked to the accuracy of SfM model (James et al., 2017b), and knowledge of the source and magnitude of error helps in interpreting the results.

 The resolution and accuracy of SfM based DEMs also relies on image quality. Low-quality images used in SfM workflow reduces the resolution and accuracy of DEMs (Russell, 2016). It has been found that image acquisition geometry affects the output of SfM models (Carbonneau and Dietrich, 2016;Morgan et al., 2017). We acknowledge that controlling image acquisition geometry in the field will be difficult as the outcrop may not be accessible from all angles for image acquisition (e.g. a boulder on steeply sloping crater wall). The error in DEMs

 depends mainly on image quality and geometry and the method of georeferencing. Proper planning of image acquisition and high GCP accuracy can improve the accuracy of DEM. Image matching is a limiting factor for point cloud density, camera calibration, error related to model scaling and orientation in the SfM workflow and DEM accuracy (Mosbrucker et al., 2017). Image matching depends on image quality, lighting condition, surface texture and the complexity of the subject. Image quality depends on good exposure (which depends on camera

 settings and lighting conditions), sharpness (i.e. the entire subject in the image in focus), noise in the image (higher ISO), camera configuration (camera sensor and lens combination). Improvement in image matching reduces reprojection error which ultimately propagates high accuracy in the dense point cloud and DEM. We have achieved a horizontal accuracy of <0.5 mm for in situ generated DEM of boulders and bedrock. To our knowledge, this accuracy has not been reported before in the literature for SfM generated DEMs of rock outcrops generated

 in the field.

### 5.1.1. DEM resolution

The resolution of the DEM depended on the resolution of camera sensor used, a distance of image acquisition from the object, quality of images and quality settings used for processing dense point cloud in Photoscan. Since

 a 24 MP camera was used and images were acquired <2 m from the boulder/bedrock resulted in DEM of resolution <1 mm/pixel. This resolution could be further increased if "Ultra High" quality settings would have been used while processing dense point cloud in Photoscan. Instead, "High" quality setting was chosen during processing dense point cloud because it cut down the time required to process DEM by 70-80% and resulted in a smaller DEM file size which can be easily handled in external analysis software (e.g. ArcGIS, Landserf).

 In our experiment, we found that "medium" dense point quality setting does not dramatically deteriorate the horizontal and vertical accuracy of DEM. The "medium" quality DEM is good enough for geomorphological studies if time and computing power are a constraint. Given optimal lighting and weather conditions, this SfM

workflow can outperform laser scanning solutions for small surfaces (<10 m$^2$). However, the performance of SfM based topographic data is affected by vegetation and shadows and texture of the surface of interest (Micheletti et al., 2015b;Smith et al., 2016).

### 5.1.2.   DEM errors

Our experiment was conducted under controlled conditions to validate sub-mm horizontal and vertical accuracy
using our triangle control target georeferencing approach. We obtained horizontal accuracy <0.60 mm and vertical accuracy of <0.45 mm in our experiment. The use of the prime lens at fixed focus will yield lower errors compared to a zoom lens for SfM photogrammetry, as suggested by Mosbrucker et al. (2017). Our experimental results suggest that prime lens had slightly better vertical accuracy compared to zoom lens when both lenses were used in autofocus mode. However, there is no statistically significant difference in the accuracy of DEMs generated
from prime and zoom lens used at autofocus. The slightly higher errors due to using zoom lens in comparison with prime lenses is acceptable considering that it offers flexibility to choose a focal length (choice depends on the field of view) and its low cost. Most of the less expensive DSLR camera lenses do not come with a focusing scale. Lens set at autofocus is more suitable than those set at the fixed focus for acquiring images of outcrops in challenging and steep terrains such as crater walls. Lens set at autofocus allows us to take sharper images from a
very close distance (few centimetres) as well as from farther away (few meters) from the rock outcrop without introducing issues associated with the hyperfocal distance of the camera system. Photoscan does an excellent job performing accurate autocalibration from EXIF data of the images. We found that using AdobeRGB colour space and tiff image compression improves the DEM accuracy. Prior Lens profile correction and the position of the control target had a negligible effect on the accuracy of DEM. Masking of images in our experiment did not reduce
the processing time for DEM generation.  We find that changing the position of the control target with respect to area of interest had an almost negligible effect on horizontal and vertical errors. For the field data, the horizontal checkpoint errors derived using rulers for Moenkopi outcrop DEMs in the field (Table 6) correspond to the results obtained in our experiment (Figure 4a). In some cases, the horizontal error was found to be lower in the field for a certain distance from the control target (Table 6) compared to the results obtained in the experiment (Figure 4a).
This could be due to better image texture of weathered outcrops in the field compared to the reduced texture of our experiment subject (Figure 3). This is evident in the reprojection error and projection accuracy (Table 4 and 5). Some of the field DEMs have lower reprojection and projection error than the DEMs generated in the experiment. Photoscan provides an option to improve the reprojection errors and thus the overall error in DEMs if errors are high due to poor image matching. This can be performed using "gradual selection" tool in Photoscan
to filter and remove tie points with high reprojection errors after image matching during stage 2 (see Table 2) of processing DEM (Agisoft, 2016).

### 5.2. Portable and affordable

For many projects, it is the budget, ease of use, and portability that require researchers to choose one technique over others. To date, relatively few studies have undertaken a cost-benefit, data acquisition rate, spatial coverage,
operating conditions, resolution and accuracy analysis of the SfM with other topography data collection methods. Some researchers (e.g. Smith et al. (2016) and Wilkinson et al. (2016)) have proposed that SfM photogrammetry ranks highly as it is the cheapest and has the highest resolution compared to other topographic data collection

methods (e.g. total station, differential GPS (dGPS), Terrestrial Laser Scanning (TLS), stereophotogrammetry). They also found that the speed of data acquisition and accuracy for SfM method is comparable to TLS and stereophotogrammetry in a close-range scenario. Our work supports their findings but goes further and outlines an approach to produce sub-mm resolution DEMs with sub-mm accuracy using ground-based, close-range SfM photogrammetry. The cost of the camera system (camera + zoom lens) used in this study is €460. The triangle control target used in this study cost less than €10. The educational licence of Agisoft Photoscan was purchased for €600 (a one time investment). The total cost of field equipment and software used in this study is well within the budget of a small research project. In addition, the total weight of the camera system and control target used is less than 1 kg. Our approach can be used in any scenario where high resolution, accurate DEMs and orthophotos are required (e.g., scaled laboratory experiments or small-scale features in the field). In addition, we have demonstrated an SfM photogrammetry approach that is relatively affordable, field-portable, fast and efficient method without requiring any prior information on camera position, orientation or internal camera parameters or the need for additional survey equipment.

### 5.3. Importance of microtopographic data in rock breakdown

We propose that the generation of microscale topographic data by methods described here will be important for the advancement of rock breakdown studies. Specific rock breakdown processes can leave a unique morphological signature on rock surfaces (Bourke and Viles, 2007). More often, the synergies linking breakdown processes, mechanisms and agents operate over a range of spatial and temporal scales (Viles, 2013) and can result in a palimpsest of features that represent a change in, e.g., weathering conditions (Ehlmann et al., 2008). As such, the breakdown is non-linear, and processes can exploit inheritance features and overprint them over time. Micro-scale DEMs will permit us to move from a predominant specific geomorphometry approach to a general geomorphometry approach, where, e.g., the relationships can be investigated. In addition, our approach will ease the cumbersome task of collecting morphometric data on individual weathering features in the field (e.g. Norwick and Dexter (2002); Bruthans et al. (2018)).

There are a number of areal surface roughness and geomorphometric parameters that can be applied to quantify rock breakdown (Leach, 2013;Lai et al., 2014;Davis et al., 2015;Du Preez, 2015;Trevisani and Rocca, 2015;Verma and Bourke, 2017). The ability to quantify surface change across an area rather than limited to specific points will aid interpretation of the causal links between controls and resultant landform development. This is particularly relevant for the recent developments in monitoring micro-climates (Mol et al., 2012;Coombes et al., 2013) of rock breakdown environments or in dynamic environments such as intertidal rock platforms (e.g., (Cullen et al., 2018)).

Our companion paper (Cullen et al., 2018) shows the potential application of our approach and provides a comparison between the traditional method of measuring erosion on rock shore platformsusing a Traversing/Micro Erosion Meter (T/MEM) with Structure from Motion (SfM) Photogrammetry. The results indicated that SfM Photogrammetry offers several advantages over the T/MEM allowing measurement of erosion at different scales on rock surfaces with low roughness while also providing a means for identifying different processes and styles of erosion.In addition the work demonstrated accuracy in the repeatability of measurements.

Diagnostic indices that reveal morphometric differences has been attempted at the landscape scale (e.g., Lyew-Ayee et al. (2007)). The production of a high-resolution dataset for microscale weathering features offers an opportunity to test analysis routines such as semi-variogram, areal surface roughness and fractal analysis to identify patterns of the breakdown features at different scales (Inkpen et al., 2000;Viles, 2001;Fardin et al., 2004;Bourke et al., 2008;Leach, 2013). Areal surface fractal analysis of rock surfaces would help to elucidate on

equifinality in the production of breakdown features and issue of distinguishing fossil from current forming features (Viles, 2001;Fardin et al., 2004)(Viles, 2001; Fardin et al., 2004).

Our approach permits the comparative study of weathering features in different environments and the same environment over time. The ability to replicate our approach to assemble a time-series of data (as outlined in a companion paper, Cullen et al. (2018), will facilitate the determination of weathering rates in the field at seasonal

and annual temporal scales. This will assist with issues in extrapolating from the laboratory to the field where rates of weathering have traditionally been overestimated (Viles, 2001).

### 6. Conclusion

We have developed and tested a triangle coded control target which is used to register SfM generated dense point

clouds to produce DEMs. We applied SfM photogrammetry on eight Moenkopi Sandstone outcrops near Meteor Crater, Arizona. We found that the deployment of existing techniques to generate high-resolution data not suitable for use in our remote and poorly accessible field terrains (e.g. crater wall, canyon). In this study, we have demonstrated that this challenge can be overcome by SfM photogrammetry. A triangle coded control target (GCPs) was specifically developed to a) compute local coordinates and b) used to georeference the 3D point cloud

generated by SfM photogrammetry. This allowed generation of a sub-mm resolution DEM with sub-mm accuracy. We validated sub-mm accuracy in DEMs with an experimental approach. Our study demonstrated that it is possible to use our method to generate DEMs of rock outcrops ($< 10$ m$^2$) in the field to sub-mm horizontal and vertical accuracy. In optimal conditions (good lighting, weather and vegetation-free) local coordinate georeferencing workflow may outperform TLS for certain applications. Development of triangle coded control

target not only helped to generate sub-mm resolution DEM but also permitted the automation of the SfM batch process workflow, generating a DEM as the end product. We anticipate that the ease of production of sub-mm resolution DEM without the use of any bulky survey equipment has the potential to transform the existing approach to small-scale topographic data acquisition and offers a promising solution to data collection challenges in the confined laboratory and difficult field conditions. The SfM workflow in this study provides an easy, simple,

quick and relatively affordable method to generate 3D topographic data for weathering features in hard to access terrains. The high-resolution DEMs of rocks surfaces in this study facilitate faster data collection and offers a potential solution to overcome many challenges in the field, including short and long-term monitoring of micro to meso-scale erosion in dynamic environments (Cullen et al., 2018).


## 5  Author contribution

A.K.V. developed the control target and designed the SfM experiment with input from M.C.B. A.K.V. collected, processed and analysed field and experimental data. A.K.V. wrote the manuscript with guidance, discussion and editing from M.C.B.

## Acknowledgement

The authors would like to thank Niamh Cullen for her assistance in the field and the review of initial version of this manuscript. The Prosser family kindly allowed authors to access the Bar T Bar Ranch property near Meteor Crater in Arizona for collecting data on Moenkopi Sandstone outcrops. A.K.V. was supported by Trinity College Dublin Postgraduate Studentship, Faculty of Engineering, Maths and Science, Trinity College Dublin, India (PhD) Scholarship, The J.N. Tata Endowment Scholarship for the higher education of Indians, and The J.N. Tata Gift Scholarship during the preparation of this manuscript. The Barringer Family Fund for Meteorite Impact Research 2015, British Society for Geomorphology Postgraduate Research Grant 2016, International Association of Sedimentologists Postgraduate Research Grant 2016, and Trinity Trust Trust Travel Grant 2016 supported A.K.V. for the fieldwork in Arizona.

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
