# Peer review of "A Structure from Motion photogrammetry-based method to generate sub-millimetre resolution Digital Elevation Models for investigating rock breakdown features"

_Earth Surface Dynamics, 2018_

## Referee Comment (RC1) · M. Schaefer (Referee) · 13 Aug 2018

Overview

The paper presented proposes a method of generating sub-millimetre accurate DEM from data collected in the field. Its main focus is the use of a triangular coded control target to scale resulting SfM models. This target is proposed as a more user-friendly, less time-consuming and cheaper alternative to other field methods. The authors present an experimental design to test the capability of SfM to generate accurate

and find optimal settings, which they then apply in the field. Their field data suggests the triangular coded control target method is successful.

Specific comments The paper discusses the difference between zoom and fixed focus lenses and finds no significant difference between them, which is interesting. It would be worth noting that the risk with zoom lenses is that the focal length can change by accident and this might affect the lens internal geometry.

The camera in the paper's experiments is set to autofocus. In theory, this could change the internal geometry of the camera between pictures, especially between the wider shots and the close range ones. Although Agisoft is now quite good at dealing with it and I have found little difference I would like to see some discussion on the topic, as Mosbrucker recommends these changes be minimised for "High Accuracy work". Were fixed focus tests done?

Table 2, was gradual selection used at all after image matching to remove tie-points with high errors? It would be worth discussing this option, as it might improve models.

In the field (4.1), what were the distance the pictures were taken? I can see from the pictures to some extend, but it would be useful to know what "all around" and "close range " mean.

In 4.1, 11, white balance is fairly immaterial for RAW images, as it can be changed in post.

In 4.1, 13-14, using autofocus does not increase depth-of-field, aperture does that. Using autofocus means the correct focal plane is chose so depth of field is optimised.

Conclusion I found the paper interesting and it contributes to the scientific discussion around SfM. It provides an actionable method for working with SfM in the field and provides some good practical advice for the different permutations of processing SfM data.

**ESurfD**

---

## Referee Comment (RC2) · P. Sapirstein (Referee) · 14 Aug 2018

The paper presents a method to create photogrammetric 3D models of rock surfaces, in support of the quantitative analysis of erosion at a micro-topographic scale (areas less than 10 m^2). The objective to capture details down to a resolution of about 0.5 mm, and controlling error within a similar scale, is met with a multistage system of field photography and software processing that involves placing a set of three coded targets in the scene. The reusable target field serves both as a scale bar and as the local coordinate system. Most of the paper is concerned with testing the accuracy of the

proposed photogrammetric method, rather than the geomorphological analysis which motivated its development.

I have waived my anonymity as a referee in part because I have published about methods to improve and assess the accuracy of photogrammetric 3D models in archaeological research. I recommend the authors consider these papers along with the additional literature cited in the bibliographies: P. Sapirstein (2018) "A high-precision photogrammetric recording system for small artifacts" J. of Cultural Heritage 31: 33–45 with 10pp suppl. P. Sapirstein, S. Murray (2017) "Establishing best practices for photogrammetry in archaeology" J. of Field Archaeology 42: 337–50 P. Sapirstein (2016) "Accurate measurement with photogrammetry at large sites," J. of Archaeological Science 66: 137–45 I enjoyed reading this paper, which has reminded me how geologists have developed interests in photogrammetry parallel to those of archaeologists, both working in similar directions. It would be good for the two areas to interact with one another more directly, such as through interdisciplinary citation and conversation.

On the positive side, the authors are to be commended for their thorough citation of geological studies involving photogrammetric modeling. Their triangular target field seems like a good, simple approach to establishing scale at remote sites where the local coordinates and north bearing need not be precisely established. The paper also includes workflows that will be useful for those wishing to learn photogrammetric recording, with many recommendations gained through practical experience.

Still, the contribution of the triangular target field is a relatively small one to the broader field of photogrammetric recording–which has witnessed an explosion in publication since the beginning of this decade about its potential and methods for its application in various contexts. More problematic is the core of the paper, a testing field that the authors use to assess accuracy of their photogrammetric method. As discussed below, the data suggest that the reference measurements on the testing field were distorted, thus invalidating any conclusions about the photogrammetric accuracy. Still, I do not doubt that their proposed method meets the requirements of the geological study, since

sub-mm resolution and accuracy is not difficult to attain with photogrammetric modeling at this scale, using the techniques they describe. I believe the paper has potential as a useful publication, but only if it is substantially reworked, beginning with fixing what must be erroneous measurements in the testing chart. Furthermore, given that the literature about photogrammetric methods is well saturated at this point, the paper also needs to make more of an effort to present an actual case study of how geological processes will be analyzed using this 3D data, which seems like the most significant potential contribution of this research.

Specific comments: A) The paper should be framed as a geological case study. It begins this way, but Section 5.3 near the end really should be near the beginning, since it is more of a proposal and justification for generating microscale topographic data in the first place. After the introduction, the paper should justify why a 0.5 or 1-mm resolution/error is needed for doing this sort of analysis, and any other attributes of the 3D models that would serve these objectives. As it stands, the choices of resolution and other processing parameters (e.g., why generate the texture at all?) come off as somewhat arbitrary, raising concerns that the workflow and processing might be needlessly complex and slow–consuming hours of human and computer time rather than minutes to generate potentially viable data. An important omission, required before the final conclusions, is an attempt to show what can be done with the 3D data, specifically the quantitative study of surface roughness, and how this was / will be carried out with the DEMs illustrated toward the end of the paper. The authors do mention a forthcoming paper about this subject, but the readers of the current paper deserve to be given a summary of the results here, and some description of the methods used to assess the 3D models / DEM data.

B) The discussion of other methods (laser scanning, MRMs) could be developed further; as it stands, there is not much basis for comparison provided (such as by laser-scanning and photographing the same subject). The authors might include more explicit estimates for the times required for these methods, at least, so the reader gets

a better notion of how photogrammetry compares practically. The software processing times with photogrammetry can be formidable, and that should be made more explicit in the comparative discussions. The paper calls the other methods "time-consuming" (eg. on page 3), but this seems rather vague, and one could easily characterize photogrammetry in the same way. I agree with the authors' assertion (page 23, section 5.2) about photogrammetry being cheap and portable, but there are important qualifications to that statement, and it has not been justified well in this paper.

More specifically, how does the roughness analysis of the photogrammetric model compare with that measured by an MRM? Even if there is not a side-by-side test, a little discussion on the resolution, accuracy, etc. of the MRM is warranted.

On page 21, the criticisms of Total station and dGPS survey seem overstated. 1) dGPS measurements of a dozen or more targets, with scale bars to fix the scale, should generate a fully georeferenced model; this is common practice in archaeology, where position and orientation are as essential as an accurate scale. I would imagine that this information would be useful in geomorphological recording as well. 2) Total station measurements are more reliable, with local errors of just a few mm, and have the option of shooting reflector-less in inaccessible locations. It would seem either piece of equipment would be advantageous in many contexts, and in fact the system proposed here with the triangular scale bar kit could be integrated for a hybrid method (e.g., placing scales and targets in the area, and measuring the coordinates with a TS). The discussion should be reframed in a more positive light to admit that these different recording methods are not mutually exclusive, but can potentially complement one another.

C) The error testing methods are problematic. First, the testing environment is nearly flat: a printed, 1.4m square printed chart with 5-cm wooden blocks set on it. The chart is also a mostly blank white sheet of paper. Not only is it completely unlike natural rock surfaces, which vary in depth and texture, the printed chart is poor for SIFT keypoint generation. The blank white background and straight black lines are not useful for

these descriptors; only the edges of the targets, the printed text, and the blocks are likely to generate reliable matches.

Second, this testing field tells us something about the accuracy of the overall scaling of the model, but not more. We are presented with no tests of the surfaces generated by the photogrammetric software (from the MVS / dense cloud stage), which might be done with repeatability tests (such as creating many models of one outcrop), or comparisons to reference data (such as created by a high resolution laser scanner). This is a significant omission, since it is the key product that is needed for assessing roughness and other parameters related to weathering. For example, poorly calibrated and oriented cameras introduce a significant amount of noise, which would make the restored 3D surface appear much rougher than the reality.

Third, while it is a good idea to separate horizontal and vertical errors, the use of two completely different testing methods (lengths of scale bars between pairs of coded targets, vs. heights of wooden blocks set on a printed sheet) means that the two error values are not comparable. How are the block heights being extrapolated in the software?

Fourth, and most troubling: figure 6a (section 3.3.3), as well as additional charts in the supplement, show a curious result that two independent photogrammetric measurements of horizontal scale bar lengths agree with one another very well, yet differ greatly (about 0.2–0.9 mm, correlated to total length) from the dimensions printed on the testing chart. That is, the consensus of 2/3 of the measurements indicate that the printed chart dimensions are incorrect. Furthermore, the discrepancies in the figures present a sawtooth pattern, flipping in the positive and negative directions (less / more than the printed chart) at a similar scale. The authors explain that they generated the testing chart dimensions in design software and printed it, presumably on a plotter, for the test photography. In their results, the dimensions for S1, S3, S6, S8, S10, S12, S14, and S16 from the two photogrammetric measures are less than the expected length on the printed chart, while the others have positive discrepancies. The explanation for

this distinctive pattern begins with the chart itself: the set I just listed are all vertically oriented on the printed sheet, while the others are horizontal.

I encourage the authors to account for this problem. Photogrammetric recording can be very precise: 1:10,000 is easily obtainable with coded targets (so, errors all below 0.1 mm in length at the size of this testing scene), and it would be difficult to conceive of how a warp that would increase scale on one axis at the expense of another could possibly be introduced. However, scaling distortions of one axis relative to another is common with printing. Some printers are in fact designed to insert subtle distortions to foil counterfeiters, but other reasons like curling of the paper (common with plotter paper) might account for these distortions–which are on the order of just 1 mm, after all, and thus would be hard to see.

Due to these problems with the chart, the conclusions about prime vs. zoom lenses, etc., are invalid, since they were tested against faulty reference measurements. If the actual lengths on the testing chart can be determined, then the photogrammetric estimates could be assessed from the same data, and the authors may be able to reproduce known phenomena in previously published research, such as improved accuracy from a fixed lens (including fixed focus settings) relative to an unstable lens.

D) On the image format (Main 2.3, Table 4, and Supplement 2.5), it is claimed that the JPEG format increases error relative to lossless formats, yet the reported increase in error is so high as to raise flags. The procedure with RAW photography converted later to TIFF adds a significant amount of time and raises storage requirements, which would only be justified if JPEG were indeed much less reliable than TIFF. In my own tests, I found a small effect, with JPEG imagery being about 97-99% as metrically consistent as TIFF images. By that, I mean repeatable for length measurements; so, for example, a TIFF-based scene with length errors of 1.00 mm might have errors of 1.02 mm if based on maximum-quality JPEGs, which for most purposes is negligible.

Of course, this could vary with processing settings and the camera. Since the text

and supplement do not specify how the JPEGs were created, it is hard to account for the very high JPEG error, but this may be due to using a relatively high compression ratio. JPEG encoding introduces strong artifacts next to high-contrast straight edges as the quality is reduced below the maximum setting; even 95% quality begins to create artifacts that could interfere with SIFT matching. Furthermore, the testing imagery is basically all black and white lines, which is exactly where JPEG performs its worst. JPEG is designed for photographs of natural forms with comparatively smooth textures, much more like the natural features in the study than the testing chart.

---

## Author Comment (AC1) · 19 Oct 2018

Response to Reviewer 1 comments RC: The paper presented proposes a method of generating sub-millimetre accurate DEM from data collected in the field. Its main focus is the use of a triangular coded control target to scale resulting SfM models. This target is proposed as a more user friendly, less time-consuming and cheaper alternative to other field methods. The authors present an experimental design to test the capability of SfM to generate accurate and find optimal settings, which they then apply in the field. Their field data suggests the triangular coded control target method is successful.

[Figure]

AC: We thank Dr Schaefer for his detailed and constructive critique of this manuscript. We appreciate the time and effort he took to review this manuscript in such detail and are pleased with the positive response. Our responses and associated changes to the manuscript are outlined below.

RC: The paper discusses the difference between zoom and fixed focus lenses and finds no significant difference between them, which is interesting. It would be worth noting that the risk with zoom lenses is that the focal length can change by accident and this might affect the lens internal geometry.

AC: We agree and have noted that during our test both the zoom lens and the prime lens were used in autofocus mode. For this work, we fixed zoom lens at 24 mm and reviewed the focal length after every certain number of images on the camera screen to make sure the focal length remained the same.

RC: The camera in the paper's experiments is set to autofocus. In theory, this could change the internal geometry of the camera between pictures, especially between the wider shots and the close range ones. Although Agisoft is now quite good at dealing with it and I have found little difference I would like to see some discussion on the topic, as Mosbrucker recommends these changes be minimised for "High Accuracy work". Were fixed focus tests done?

AC: We did not perform fixed focus tests as the images were taken in autofocus mode at our field site in Arizona. We aimed to validate the accuracy of the DEMs generated from field data through an experimental approach. In a challenging and steep terrain such as impact crater wall where there is not much space around an outcrop to take images from all aspects (see Fig 1). The autofocus mode enables us to get sharp images from wider and close range shots. We discuss this in section 5.1.2, line 7-17, page 24.

RC: Table 2, was gradual selection used at all after image matching to remove tie-points with high errors? It would be worth discussing this option, as it might improve

models.

AC: The errors in our DEMs were low. So, we did not use the gradual selection option to remove points with higher reprojection errors, but we agree that this option can be used in case of high reprojection errors. We now include this in section 5.1.2, line 29-32, page 24.

RC: In the field (4.1), what were the distance the pictures were taken? I can see from the pictures to some extend, but it would be useful to know what "all around" and "close range " mean.

AC: We have inserted approximate image acquisition distances in paragraph 2 (section 4.1, line 14-15, page 16). RC: In 4.1, 11, white balance is fairly immaterial for RAW images, as it can be changed in post.

AC: We agree but retain it in the manuscript for researchers who may use in camera-generated Jpeg images due to storage constraint, and in this case, the white balance settings can be significant.

RC: In 4.1, 13-14, using autofocus does not increase depth-of-field, aperture does that. Using autofocus means the correct focal plane is chose so depth of field is optimised.

AC: We agreed and have modified the sentence in section 4.1, line 11, page 16.

RC: Conclusion I found the paper interesting and it contributes to the scientific discussion around SfM. It provides an actionable method for working with SfM in the field and provides some good practical advice for the different permutations of processing SfM data.

Response to Reviewer 2 comments

RC: The paper presents a method to create photogrammetric 3D models of rock surfaces, in support of the quantitative analysis of erosion at a micro-topographic scale (areas less than 10 mËĘ2). The objective to capture details down to a resolution of

about 0.5 mm, and controlling error within a similar scale, is met with a multistage system of field photography and software processing that involves placing a set of three coded targets in the scene. The reusable target field serves both as a scale bar and as the local coordinate system. Most of the paper is concerned with testing the accuracy of the proposed photogrammetric method, rather than the geomorphological analysis which motivated its development.

AC: We are grateful for the detailed and constructive criticism provided by Dr Sapirstein, and we appreciate the time he took to review this manuscript. Responses to comments and suggested edits are outlined below.

RC: I have waived my anonymity as a referee in part because I have published about methods to improve and assess the accuracy of photogrammetric 3D models in archaeological research. I recommend the authors consider these papers along with the additional literature cited in the bibliographies: P. Sapirstein (2018) "A high-precision photogrammetric recording system for small artifacts" J. of Cultural Heritage 31: 33–45 with 10pp suppl. P. Sapirstein, S. Murray (2017) "Establishing best practices for photogrammetry in archaeology" J. of Field Archaeology 42: 337–50 P. Sapirstein (2016) "Accurate measurement with photogrammetry at large sites," J. of Archaeological Science 66: 137–45 I enjoyed reading this paper, which has reminded me how geologists have developed interests in photogrammetry parallel to those of archaeologists, both working in similar directions. It would be good for the two areas to interact with one another more directly, such as through interdisciplinary citation and conversation.

AC: We thank Dr Sapirstein for recommending additional literature relevant to the SfM technique. We found these papers very useful, and they are cited in section 1.1, line 6-8, page 4 and in section 2.1, line 13, page 6. RC: On the positive side, the authors are to be commended for their thorough citation of geological studies involving photogrammetric modeling. Their triangular target field seems like a good, simple approach to establishing scale at remote sites where the local coordinates and north bearing need not be precisely established. The paper also includes workflows that will be useful for

**ESurfD**
those wishing to learn photogrammetric recording, with many recommendations gained through practical experience. Still, the contribution of the triangular target field is a relatively small one to the broader field of photogrammetric recording–which has witnessed an explosion in publication since the beginning of this decade about its potential and methods for its application in various contexts.

AC: We agree that there has been a considerable amount of published work that demonstrates the use of SfM for production of high-resolution topographic data. However, to our knowledge, there has been no study that has produced DEM of field rock surfaces with mm to sub-mm accuracy in steep and limited access terrains (e.g. steep canyon wall and mountain slope, and impact crater wall shown in Fig 1).

Fig 1. USGS slope map of Meteor Crater. Crater walls are very steep ranging from 40° to 82°. The steep and unstable crater walls (due to loose ejecta fragments) makes it very challenging to acquire micro-topographic data on outcrops.

We disagree with the reviewer on the contribution of the triangular control target. We feel a significant contribution of this work is the development of a field-portable control target and the demonstration of its efficacy in difficult field terrains. A review of the literature suggests that the potential of the SfM technique to study breakdown features at mm-cm scale has not been adequately explored by researchers in rock weathering studies . An intention of this manuscript is to bridge this gap and advance data collection methods for mm-cm scale weathering features.

RC: More problematic is the core of the paper, a testing field that the authors use to assess accuracy of their photogrammetric method. As discussed below, the data suggest that the reference measurements on the testing field were distorted, thus invalidating any conclusions about the photogrammetric accuracy. AC: We understand the concerns of Dr Sapirstein regarding our testing chart approach for error evaluation. We would like to emphasise that in order to compute the error propagation with distance, it was necessary to fix the checkpoints and triangle GCP system with respect

to each other. This was done to estimate the distance between checkpoints from the location of the triangle control target. In our opinion, it is difficult to determine the distance between checkpoints and triangle control target with an accuracy of less than ≤1 mm using measuring tape in the field. Specific comments related to error evaluation experiment are addressed below.

RC: Still, I do not doubt that their proposed method meets the requirements of the geological study, since sub-mm resolution and accuracy is not difficult to attain with photogrammetric modelling at this scale, using the techniques they describe. I believe the paper has potential as a useful publication, but only if it is substantially reworked, beginning with fixing what must be erroneous measurements in the testing chart.

AC: See above.

RC: Furthermore, given that the literature about photogrammetric methods is well saturated at this point, the paper also needs to make more of an effort to present an actual case study of how geological processes will be analyzed using this 3D data, which seems like the most significant potential contribution of this research.

AC: A case study on the application of the approach developed is beyond the scope of this submission. Indeed, we bring to the editor's attention the recent publication of a companion paper in ESurf Cullen, N. D., Verma, A. K., and Bourke, M. C.: A comparison of Structure from Motion Photogrammetry and the Traversing Micro Erosion Meter for measuring erosion on rock shore platforms, Earth Surf. Dynam. Discuss., https://doi.org/10.5194/esurf-2018-552018. The latter is an application of the technique to experimental rock shore platform erosion.

RC: Specific comments: A) The paper should be framed as a geological case study. It begins this way, but Section 5.3 near the end really should be near the beginning, since it is more of a proposal and justification for generating microscale topographic data in the first place. After the introduction, the paper should justify why a 0.5 or 1-mm resolution/ error is needed for doing this sort of analysis, and any other attributes of the

3D models that would serve these objectives. As it stands, the choices of resolution and other processing parameters (e.g., why generate the texture at all?) come off as somewhat arbitrary, raising concerns that the workflow and processing might be needlessly complex and slow–consuming hours of human and computer time rather than minutes to generate potentially viable data.

AC: We agree and have revised the introduction to emphasise the importance of sub-mm accuracy/resolution DEMs in rock breakdown studies (section 1, line 7-22, page 2). We suggest that the availability of a high resolution-low error (sub-mm) DTM is important at the small scale (mm-cm) for weathering features. Generating textures help to generate more precise and detailed orthomosaic of the rock surface. This is a value-added product for analysis. For example, the orthomosaic can be used to map the abundance/frequency of weathering features. It can also be used to identify additional signatures of weathering such as tone, colour which may be used to identify lichen, weathering rind, patina, dust adhesion etc. Further, we have edited the introduction to clarify the importance of developing a cross-scalar technique for analysis of rock breakdown features. We have retained the more appropriate parts of section 5.3 in our discussion to emphasise the impact of our approach.

RC: An important omission, required before the final conclusions, is an attempt to show what can be done with the 3D data, specifically the quantitative study of surface roughness, and how this was / will be carried out with the DEMs illustrated toward the end of the paper. The authors do mention a forthcoming paper about this subject, but the readers of the current paper deserve to be given a summary of the results here, and some description of the methods used to assess the 3D models / DEM data.

AC: We respectfully disagree. We assert first that it is important to demonstrate and test the efficacy and accuracy of a newly developed triangular coordinate system. The generation of sub-mm scale DEMs of rock surfaces using SfM requires detailed treatment in a stand-alone paper. Second, we contend that this is required to be reviewed and approved prior to the demonstration of the application in the study of rock weathering and erosion analysis. This is why our paper is a companion paper to Cullen et al. (2018). In that paper, we apply our method to study erosion on simulated rock shore platforms. We have added a summary of some of the findings in Cullen et al., 2018 in section 5.3, line 32-37, page 25.

RC: B) The discussion of other methods (laser scanning, MRMs) could be developed further; as it stands, there is not much basis for comparison provided (such as by laserscanning and photographing the same subject). The authors might include more explicit estimates for the times required for these methods, at least, so the reader gets a better notion of how photogrammetry compares practically. The software processing times with photogrammetry can be formidable, and that should be made more explicit in the comparative discussions. The paper calls the other methods "time-consuming" (eg. on page 3), but this seems rather vague, and one could easily characterize photogrammetry in the same way.

AC: We agree and have re-written the paragraph to avoid vague comparisons between SfM and laser scanning. We have also cited literature which readers can refer for a detailed comparison between TLS and SfM techniques (section 1.1, line 21-23, page 4).

RC: I agree with the authors' assertion (page 23, section 5.2) about photogrammetry being cheap and portable, but there are important qualifications to that statement, and it has not been justified well in this paper.

AC: We modified section 5.2 and added the cost and weight of the equipment in the discussion to justify this point in the paper (section 5.2, line 5-9, page 25).

RC: More specifically, how does the roughness analysis of the photogrammetric model compare with that measured by an MRM? Even if there is not a side-by-side test, a little discussion on the resolution, accuracy, etc. of the MRM is warranted.

AC: A detailed roughness analysis of rock surfaces is beyond the scope of this paper.

However, we have included a new, short discussion of the advantages and constraints of roughness estimates using these two techniques (section 1, line 16-23, page 3).

RC: On page 21, the criticisms of Total station and dGPS survey seem overstated.

AC: The efficacy of total station and dGPS equipment are limited in certain landscapes. In our experience, limitations of dGPS include loss of signal near a cliff, steep canyon or a crater wall. As the aim of this study is to develop a portable method for deployment in challenging terrains such as impact craters or mountain slopes where heavy survey equipment are not suitable. For terrains where these instruments have limited use, we find it appropriate to highlight those specific limitations. While other studies were successful in generating DEMs with $\leq\sim1$ mm accuracy, they all required survey equipment (e.g. total station, laser range finder) for validation (section 5, page 22-23). Our technique provides an alternative approach.

RC: 1)dGPS measurements of a dozen or more targets, with scale bars to fix the scale, should generate a fully georeferenced model; this is common practice in archaeology, where position and orientation are as essential as an accurate scale. I would imagine that this information would be useful in geomorphological recording as well.

AC: We agree; however, we note that the reviewer refers to a measurement scale that is larger than that under discussion. The accurate reconstruction of position and orientation of m scale outcrop/boulder is not as relevant for the study of small-scale (mm-cm) weathering features. What may be relevant is the orientation of orthophoto and DEMs for further analysis. For example, using geographic coordinate for an outcrop surface dipping 90° creates DEM and orthophoto oriented vertically which hinders identification of small-scale weathering features. Whereas our approach of keeping the triangle control target parallel to the outcrop surface enables us to build DEMs and Orthophotos in the XY plane (horizontal) regardless of the orientation of outcrop in the field. While, this can be achieved in post-processing by rotating the model, it makes DEM generation more complicated.

RC: 2) Total station measurements are more reliable, with local errors of just a few mm, and have the option of shooting reflector-less in inaccessible locations. It would seem either piece of equipment would be advantageous in many contexts, and in fact the system proposed here with the triangular scale bar kit could be integrated for a hybrid method (e.g., placing scales and targets in the area, and measuring the coordinates with a TS). The discussion should be reframed in a more positive light to admit that these different recording methods are not mutually exclusive but can potentially complement one another.

AC: We agree that total station measurements are reliable and accurate to few mm and have the option of shooting reflector-less inaccessible locations. There are many studies which have used the total station to calculate local coordinates from GCPs, but none have used it in challenging terrains to create sub-mm accuracy DEMs (section 5, page 22-23). Total station is an expensive piece of equipment, and our aim in this study is to produce the highest accuracy DEMs without the requirement of expensive and bulky survey equipment and we have made that clear in the paper. We highlight again that a Total Station requires a stable tripod to operate at highest accuracy. On steep, unstable crater walls it is not possible to reliably deploy a tripod to operate a TS (see Fig 1).

RC: C) The error testing methods are problematic. First, the testing environment is nearly flat: a printed, 1.4m square printed chart with 5-cm wooden blocks set on it. The chart is also a mostly blank white sheet of paper. Not only is it completely unlike natural rock surfaces, which vary in depth and texture, the printed chart is poor for SIFT keypoint generation. The blank white background and straight black lines are not useful for these descriptors; only the edges of the targets, the printed text, and the blocks are likely to generate reliable matches.

AC: We agree that the flat white chart is not the ideal subject for image matching during sparse point cloud generation. That is why we also imaged then ground surface made up of paving stones near the error evaluation chart. We secured good matches for the

**ESurfD**
[Figure]

black lines, texts, wooden blocks and nearby ground surface. We found that tie points generated for the test chart and nearby ground surface are comparable to similar area rock surface. When we processed sparse point clouds to the dense point cloud, we observe no holes in our model (see Fig 2). We were able to generate high-resolution DEM from the dense point cloud.

Fig 2. Dense point cloud screen shot from Agisoft Photoscan (zoomed in view).

RC: Second, this testing field tells us something about the accuracy of the overall scaling of the model, but not more. We are presented with no tests of the surfaces generated by the photogrammetric software (from the MVS / dense cloud stage), which might be done with repeatability tests (such as creating many models of one outcrop), or comparisons to reference data (such as created by a high resolution laser scanner). This is a significant omission, since it is the key product that is needed for assessing roughness and other parameters related to weathering. For example, poorly calibrated and oriented cameras introduce a significant amount of noise, which would make the restored 3D surface appear much rougher than the reality.

AC: We agree that tests using the error evaluation chart tell us about the overall accuracy of the model generated using our control target approach. We now included a DEM of Difference (DoD) which are the results of DEMs generated using two independent set of images of the same surface (section 3.3.3, line 20-38, page 14-15). In our companion paper, we have focussed on demonstrating the repeatability of the approach on a small scale ($\sim$100 cm2) simulated rock platform surfaces.

RC: Third, while it is a good idea to separate horizontal and vertical errors, the use of two completely different testing methods (lengths of scale bars between pairs of coded targets, vs. heights of wooden blocks set on a printed sheet) means that the two error values are not comparable. How are the block heights being extrapolated in the software? AC: We disagree and suggest that it is a robust approach. We propose that it is appropriate to separate horizontal and vertical errors as they represent different

planes, i.e. xy and z. We derived the horizontal and vertical errors from two separate set of checkpoints (lengths of scale bars between pairs of coded targets, vs. heights of wooden blocks set on a printed sheet). In addition, we have shown that horizontal and vertical errors of DEMs can be compared separately between DEMs. The block height was estimated using the 3D Analyst tool in ArcMap, detail of this is outlined in section 3.2, line 12-20, page 11-12.

RC: Fourth, and most troubling: figure 6a (section 3.3.3), as well as additional charts in the supplement show a curious result that two independent photogrammetric measurements of horizontal scale bar lengths agree with one another very well, yet differ greatly (about 0.2-0.9 mm, correlated to total length) from the dimensions printed on the testing chart. That is, the consensus of 2/3 of the measurements indicate that the printed chart dimensions are incorrect. Furthermore, the discrepancies in the figures present a sawtooth pattern, flipping in the positive and negative directions (less / more than the printed chart) at a similar scale. The authors explain that they generated the testing chart dimensions in design software and printed it, presumably on a plotter, for the test photography. In their results, the dimensions for S1, S3, S6, S8, S10, S12, S14, and S16 from the two photogrammetric measures are less than the expected length on the printed chart, while the others have positive discrepancies. The explanation for this distinctive pattern begins with the chart itself: the set I just listed are all vertically oriented on the printed sheet, while the others are horizontal. I encourage the authors to account for this problem. Photogrammetric recording can be very precise: 1:10,000 is easily obtainable with coded targets (so, errors all below 0.1 mm in length at the size of this testing scene), and it would be difficult to conceive of how a warp that would increase scale on one axis at the expense of another could possibly be introduced. However, scaling distortions of one axis relative to another is common with printing. Some printers are in fact designed to insert subtle distortions to foil counterfeiters, but other reasons like curling of the paper (common with plotter paper) might account for these distortions–which are on the order of just 1 mm, after all, and thus would be hard to see.

AC: We agree and thank the reviewer for highlighting out the pattern of horizontal errors along x and y-axis. We tried to obtain the highest quality print of testing chart on a 200-gsm high-quality satin material from a professional printing service. We would assume that there is the minimum possible or no distortion due to printing. We did not obtain this sharp sawtooth pattern for the vertical errors which were determined from fixed wooden blocks on the test chart. The sawtooth pattern in the figures for horizontal errors can be explained due to a slight curl in the paper while laying it on the ground. However, since the chart is white, the curl was not visible during the experiment or in the orthophoto. Although during the close examination of high-resolution DEM (Fig 3), the curling of paper can be seen which explains the sawtooth pattern of errors and negative errors along the x-axis and positive errors along the y-axis in Figure 4 a and c. Since the chart was the test chart was fixed with tape on the ground, this should not be an issue in independent image surveys and comparison of the DEMs.

Fig 3. DEM of the chart area. Slight curls can be observed in the chart.

RC: Due to these problems with the chart, the conclusions about prime vs. zoom lenses, etc., are invalid, since they were tested against faulty reference measurements. If the actual lengths on the testing chart can be determined, then the photogrammetric estimates could be assessed from the same data, and the authors may be able to reproduce known phenomena in previously published research, such as improved accuracy from a fixed lens (including fixed focus settings) relative to an unstable lens.

AC: We used a zoom lens and prime lens set at autofocus mode. Our evaluation chart was fixed to the ground and wooden blocks were fixed to the chart. We determined the actual height of the wooden blocks using digital callipers and later subtracted it from the measured height from the DEM to calculate the error. Since the wooden blocks are solid and not affected by external factors, we hold that the vertical error obtained from them are accurate and thus can be used to compare the performance of the lenses. In addition, we contend that although the test chart was slightly curled, it was fixed to the ground. We imaged the same surface in three independent surveys producing

different DEMs. Therefore, in theory, we should be able to compare these DEMs using horizontal errors calculated from the test chart. We did find that prime lens performed slightly better in some tests but there was no statistically significant difference found between the zoom and prime lens performance when set on autofocus. We agree that prime lens set at fixed focus might perform better than a zoom lens. However, using lens set at fixed focus was not suitable in field conditions. So, we did not test the prime lens at fixed focus.

RC: D) On the image format (Main 2.3, Table 4, and Supplement 2.5), it is claimed that the JPEG format increases error relative to lossless formats, yet the reported increase in error is so high as to raise flags. The procedure with RAW photography converted later to TIFF adds a significant amount of time and raises storage requirements, which would only be justified if JPEG were indeed much less reliable than TIFF. In my own tests, I found a small effect, with JPEG imagery being about 97-99% as metrically consistent as TIFF images. By that, I mean repeatable for length measurements; so, for example, a TIFF-based scene with length errors of 1.00 mm might have errors of 1.02 mm if based on maximum-quality JPEGs, which for most purposes is negligible. Of course, this could vary with processing settings and the camera. Since the text and supplement do not specify how the JPEGs were created, it is hard to account for the very high JPEG error, but this may be due to using a relatively high compression ratio. JPEG encoding introduces strong artifacts next to high-contrast straight edges as the quality is reduced below the maximum setting; even 95% quality begins to create artifacts that could interfere with SIFT matching. Furthermore, the testing imagery is basically all black and white lines, which is exactly where JPEG performs its worst. JPEG is designed for photographs of natural forms with comparatively smooth textures, much more like the natural features in the study than the testing chart.

AC: We agree that JPEG performed worst for smooth textured and flat evaluation chart which is evident in relatively high reprojection error. Our result shows slightly higher horizontal checkpoint errors (RMSE = 0.91 mm) compared to tiff format (RMSE = 0.59

mm). Whereas vertical errors for jpeg format were similar to tiff format. In our opinion, this difference in errors is acceptable if storage is a constraint. The difference in the JPEG error could be explained by in-camera JPEG compression. We used fine JPEG with medium image size on Nikon D5500.

Again, we would like to thank the Reviewers for the thorough review and highlighting important challenges that enabled us to improve our approach.

[Figure]

[Figure]

**Fig. 1.** USGS slope map of Meteor Crater. Crater walls are very steep ranging from 40° to 82°. The steep and unstable crater walls (due to loose ejecta fragments) makes it very challenging to acquire micro-top

![Perspective 3D point cloud screenshot showing markers on a white square surface. points: 47,933,773]

**Fig. 2.** Dense point cloud screen shot from Agisoft Photoscan (zoomed in view).

**Fig. 3.** DEM of the chart area. Slight curls can be observed in the chart.